



# Oceanic primary production decline halved in eddy-resolving simulations of global warming

Damien Couespel[1], Marina Lévy[1], and Laurent Bopp[2]

[1]Sorbonne Université, LOCEAN-IPSL, CNRS/IRD/MNHN, Paris, France
[2]LMD-IPSL, École Normale Supérieure/PSL University, CNRS, École Polytechnique, Sorbonne Université, Paris, France
**Correspondence:** Damien Couespel (damien.couespel@locean-ipsl.upmc.fr)

**Abstract.** The decline in ocean primary production is one of the most alarming consequences of anthropogenic climate change. This decline could indeed lead to a decrease in marine biomass and fish catch, as highlighted by recent policy-relevant reports. Because of computational constraints, current Earth System Models used to project ocean primary production under global warming scenarios have to parameterize flows occurring below the resolution of their computational grid (typically 1°). To

overcome these computational constraints, we use an ocean biogeochemical model in an idealized configuration representing a mid-latitude double-gyre circulation, and perform global warming simulations under increasing horizontal resolution (from 1° to 1/27°) and under a large range of parameter values for the eddy parameterization employed in the coarse resolution configuration. In line with projections from Earth System Models, all our simulations project a marked decline in net primary production in response to the global warming forcing. Whereas this decline is only weakly sensitive to the eddy parameters

in the eddy-parameterized coarse resolution, the simulated decline in primary production in the subpolar gyre is halved at the finest eddy-resolving resolution (−12 % at 1/27° vs −26 % at 1° at the end of the 70 years long global warming simulations). This difference stems from the high sensitivity of the sub-surface nutrient transport to model resolution. Our results call for improved representation of the role of eddies on nutrient transport below the seasonal mixed-layer to better constrain the future evolution of marine biomass and fish catch potential.

## 1 Introduction

A decrease in global marine animal biomass and in fisheries catch potential is projected over the 21st century under all emission scenarios and this decrease is mostly driven by the projected decline in phytoplankton primary production in response to anthropogenic climate change (Bindoff et al., 2019). Phytoplankton are microorganisms essential to the earth system through their influence on the global carbon cycle and biological sequestration of atmospheric $CO_2$ (Volk and Hoffert, 1985; Falkowski

et al., 1998; Field et al., 1998). They are also at the base of most oceanic food webs and as such ultimately constrain fish biomass and fish catch potential in the ocean (Pauly and Christensen, 1995; Chassot et al., 2010; Friedland et al., 2012; Stock et al., 2017). For these reasons, Net Primary Production (NPP) projections are at the heart of policy relevant reports (IPCC, 2019; IPBES, 2019), whose conclusions on the critical role of the microbial world for addressing climate change and





accomplishing the United Nations Sustainable Development Goals have been recently reaffirmed in a "Scientists' warning to
humanity" consensus statement (Cavicchioli et al., 2019).

Earth System Models (ESMs) have become essential tools for projecting how greenhouse gas emissions will affect global biogeochemical cycling in the future (Bonan and Doney, 2018). Model mean ESMs projections show that under a wide range of emission scenarios, global ocean NPP will decline in the 21st century and beyond, principally due to the reduced supply of inorganic nutrients from the sub-surface where they are abundant to the sunlit ocean where phytoplankton photosynthesis
occurs (Steinacher et al., 2010; Bopp et al., 2013; Moore et al., 2018; Bindoff et al., 2019; Kwiatkowski et al., 2020). Due to strong computational constraints, an important limitation of ESMs is that transient eddies and jet like flows of horizontal scales 100 km and less fall below the size of their computational ocean grid and are crudely parameterized (Gent and McWilliams, 1990). Eddy-parametrized coarse resolution (1° or coarser) ESMs capture the main contrasts in nutrient distribution at the global scale (Séférian et al., 2020). But they fail to capture the full complexity of these distributions at the regional scale
related with the action of eddies and fine scale flows (McGillicuddy Jr., 2016; Mahadevan, 2016; Lévy et al., 2018), despite ongoing effort in the development of parametrizations (Fox-Kemper et al., 2008).

Current projections of NPP decline are based on eddy-parameterized models and thus are potentially biased by their inadequate representation of sub-grid processes (Bahl et al., 2020). Here, we assess the difference that eddy resolution makes to shaping the response of NPP to future global warming. To circumvent the computational constrains, we use a biophysical
model in a size reduced setting representing a double-gyre circulation (Fig. 1). Such a configuration is a crude approximation of the mid-latitude North Atlantic or North Pacific circulations comprising subpolar and subtropical gyres separated by the Gulf Stream or the Kuroshoïo. We perform transient climate change simulations under an idealized global warming scenario in the form of a linear increase in atmospheric temperature, corresponding to a typical high carbon-emission scenario in the 21st century (see the Methodology section and Fig. 1). By comparing the response to the same scenario for five eddy-parameterized
coarse resolution (1°) and two eddy-resolving fine resolution (1/9° and 1/27°) model configurations (Fig. 1 and 2), we show that the projected NPP decline under global warming is halved with the eddy-resolving configurations.

The traditional paradigm attributes declining NPP to increased vertical stratification that diminishes the vertical supply of nutrients by diapycnal mixing (Doney, 2006; Steinacher et al., 2010; Fu et al., 2016; Bindoff et al., 2019; Kwiatkowski et al., 2020). More recent understanding has shed light on the significant contribution of lateral advective supplies (Palter et al., 2005;
Lévy, 2005; Lozier et al., 2011; Letscher et al., 2016; McKinley et al., 2018). This lateral nutrient transport is known as the nutrient stream (Williams et al., 2006; Palter and Lozier, 2008; Williams et al., 2011), a strong advective nutrient flux carried in sub-surface waters and associated with Western Boundary Currents (WBCs) and Meridional Overturning Circulation (MOC). Recent results based on ESM projections have suggested that the weakening of the MOC under global warming would reduce the nutrient stream and act in concert with the reduced vertical mixing to induce a decline in NPP (Whitt, 2019; Whitt and
Jansen, 2020; Tagklis et al., 2020). An explicit resolution of eddies in ocean models is known to change the position of WBCs (Chassignet and Marshall, 2008; Lévy et al., 2010; Chassignet and Xu, 2017), the strength of the MOC (Hirschi et al., 2020), and to increase both stratification (Chanut et al., 2008; Lévy et al., 2010; Karleskind et al., 2011) and the advective supply of





nutrients (Lévy et al., 2012a; Uchiyama et al., 2017; Uchida et al., 2020). But these effects from resolving eddies have not yet been explored in global warming scenarios, and not to our knowledge not in terms of their impacts on projected NPP.

## 2   Methodology

To overcome the computational constraints associated with the challenge of running eddy-resolving simulations of climate change we adopt an idealized framework using a size reduced setting configuration with an idealized scenario of global warming. To investigate the processes driving the decline in NPP we base our analysis on nitrate budgets. The following sections describe 1) the experimental design, 2) the mean equilibrium states and 3) the diagnostics used in this study.

### 2.1   Models, configurations, simulations and experimental design

Ocean physics are simulated using the primitive equation ocean model NEMO (Madec et al., 2017). Density is defined from salinity and temperature using a bilinear state equation. Vertical mixing is computed with a turbulent kinetic energy closure scheme, with a background value of $10^{-5}$ $\mathrm{m^2.s^{-1}}$ and local enhancement to $100$ $\mathrm{m^2.s^{-1}}$ in case of convection (unstable density profile). For biogeochemistry, we used the LOBSTER model (Lévy et al., 2005, 2012b), implemented in NEMO, which uses nitrogen as a currency and computes the evolution of 6 tracers: phytoplankton, zooplankton, dissolved organic material, nitrate, ammonium and sinking detritus.

The model is run in a configuration adapted from previous studies (Krémeur et al., 2009; Lévy et al., 2012b; Resplandy et al., 2019). The domain geometry is a closed square basin on the $\beta$-plane centred at $\sim 30\,°$N, 3180 km wide and long, and 4 km deep. The domain is bounded by vertical walls and by a flat bottom with a free slip boundary condition. The sea surface is kept free. The circulation is forced by analytical zonal distributions of wind stress, net heat flux and fresh water flux. The wind and buoyancy forcings are such that a double-gyre circulation is set up (Fig. 1 and A2). The wind stress is zonal and varies latitudinally in a sinusoidal manner between the extrema at the edges and the middle of the domain. The net heat flux takes the form of a restoring toward a zonal apparent air temperature profile. A portion of the net heat flux comes from solar radiation and is allowed to penetrate within the water column. A fresh water flux is also prescribed and varies zonally. It is determined such as, at each time step, the basin-integrated flux is zero.

We use 3 horizontal resolutions: 106 km (1°), 12 km (1/9°) and 4 km (1/27°). For each resolution, time steps, numerical schemes and lateral diffusion are adapted (see table 1). On the vertical, 30 z-levels are used whose thickness varies from 10 metres at the surface to 500 metres at depth. The upper 120 metres contain 10 vertical levels, the upper 500 metres contain 20 vertical levels. For the 1° resolution configurations, we used the Gent and McWilliams (1990) (GM) eddy parameterization. This parameterization relies on two coefficients, an isopycnal turbulent Redi coefficient (hereafter, lateral diffusion coefficient) and a GM coefficient that translates into a bolus velocity that acts to flatten isopycnals. For testing the sensitivity to the Gent and McWilliams (1990) parameterization, we used 5 combinations of the lateral diffusion and GM coefficients: 1) 500 $\mathrm{m^2.s^{-1}}$, 2) 1000 $\mathrm{m^2.s^{-1}}$ and 3) 2000 $\mathrm{m^2.s^{-1}}$ for both parameters, and 4) 500 $\mathrm{m^2.s^{-1}}$ and 5) 2000 $\mathrm{m^2.s^{-1}}$ for the lateral diffusion parameter but keeping the GM coefficient at 1000 $\mathrm{m^2.s^{-1}}$. We thus end up with 7 different configurations: 5 eddy-



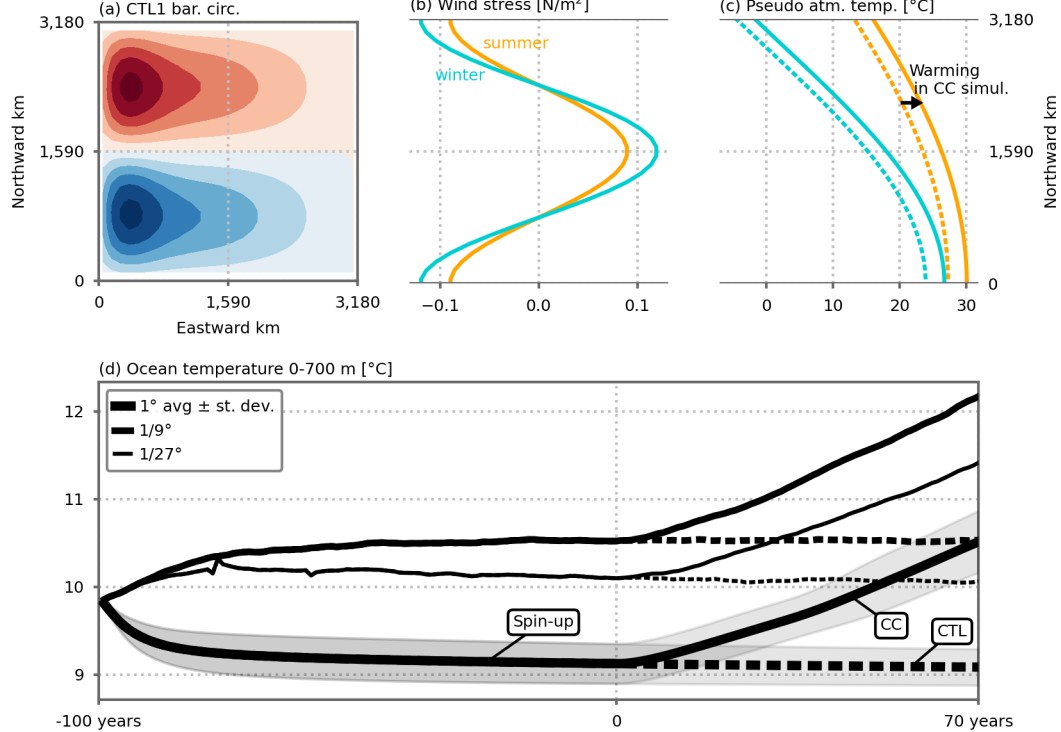

**Figure 1.** (a) Barotropic circulation over the model domain (average of the five 1° resolution preindustrial control simulations, see also Fig. A2 for individual configurations) and analytical external forcings as a function of latitude: (b) wind stress (N.m²), and (c) pseudo atmospheric temperature (°C). The forcings vary seasonally in a sinusoidal manner between winter (blue) and summer (orange) extrema. In the climate change simulations, the pseudo atmospheric temperature increases linearly in time between the dashed and solid lines (panel c). After 70 years of simulation, the linear atmospheric temperature increase reaches +2.8 °C. (d) Time series of annual ocean temperature (°C) averaged between 0 and 700 metres depth over the model domain. Years −100 to 0 correspond to the spin-up period under seasonal preindustrial forcing. The preindustrial control and climate change simulations discussed in this paper start at year 0. The 1° resolution time series shows the average of the five 1° configurations. Shading indicates ± one standard deviation. See Fig. A1 for individual 1° model configurations. The small peaks in the time series of the 1/27° resolution spin-up around years −80 and −60 are due to a problem during the chaining of the spin-up jobs (which were set by periods of 2 years). These errors did not influence the equilibrium states and we decided not to rerun the spin-up simulations to save computational resources.

parameterized at coarse resolution (1°) and 2 eddy-resolving at fine resolution (1/9° and 1/27°). In the following, results from the eddy-parameterized coarse resolution configurations are synthesized by showing the average ± one standard deviation across the 5 different configurations. Contrary to the 1/27° configuration, the qualifier "eddy-permitting" is probably more appropriate for the 1/9° configuration. Nevertheless to simplify and as the emphasis is put on the differences between the 1° resolution and the finer ones we use the term eddy-resolving for both.





|  | CTL1/CC1 | CTL9/CC9 | CTL27/CC27 |
|---|---|---|---|
| **Horizontal resolution** | 1° or 106 km | 1/9° or 11.8 km | 1/27° or 3.9 km |
| **Grid cell number** | $30 \times 30$ | $270 \times 270$ | $810 \times 810$ |
| **Time step** | 30 min | 16 min | 5 to 4.5 min |
| **Number of CPU** | 1 | 64 | 800 |
| **CPU time (cpu. × real time)** | 20 min./simulated year | 260 h./simulated year | 7000 h./simulated year |
| **Momentum diffusion** | Horizontal laplacian | None | None |
| **Viscosity coef.** | $10^5$ m$^2$.s$^{-1}$ | None | None |
| **Tracer advection scheme** | TVD | MUSCL | MUSCL |
| **Tracer diffusion** | Isopycnal laplacian | None | Horizontal bilaplacian |
| **k$_{redi}$ diffusion coef.** | 500, 1000, 2000 m$^2$.s$^{-1}$ | None | $-10^9$ m$^4$.s |
| **k$_{gm}$ GM coef.** | 500, 1000, 2000 m$^2$.s$^{-1}$ | None | None |

**Table 1.** Resolution-dependent model features and parameters. For details on numerical schemes see Madec et al. (2017) and Lévy et al. (2001). Note that we added a minimum level of bilaplacian tracer diffusivity at 1/27° to insure numerical stability, and this was not needed at 1/9° resolution.

The basin is initialised at rest with vertical profiles of temperature and salinity uniformly applied to the whole domain and homogeneous concentrations of the biogeochemical tracers. A 2000 years spin-up with a eddy-parameterized coarse resolution configuration (1°, k$_{gm}$=k$_{redi}$=1000 m$^2$.s$^{-1}$) is then performed under preindustrial forcings (wind, air temperature and fresh water flux) varying seasonally in a sinusoidal manner between winter and summer extrema and repeated every year. Then, for each configuration, a climate change and a preindustrial control simulations are carried out from this initial coarse resolution
spin-up state in 2 steps (Fig. 1 and A1):

1. 100 years of spin-up under the same preindustrial control forcings, in order to reach an equilibrium for each configuration. These seven additional spin-up (one for each configuration) are initialized with the annual mean coarse resolution spin-up state. Notable is that the final dynamical and biogeochemical equilibrium are different for each configuration (Fig. 1, 2, A1, A2, A4). The following section briefly describes these equilibrium and their differences.

2. 70 years simulation for each configuration, following 2 scenarios. The preindustrial control scenario is the continuation of the spin-up, forcings are kept seasonal. In the climate change scenario, for air temperature, in addition to seasonal variations, we impose a linear trend of +0.04 °C per year ($\sim$ +2.8 °C after 70 years). This trend roughly corresponds to the trend in surface air temperature of the simulations following the SSP5-8.5 scenario (Tokarska et al., 2020). In total, 14 simulations of 70 years were run: 7 configurations × 2 scenarios. In the following we will compare the differences between the eddy-resolving simulations (1/9° and 1/27° resolutions) and the mean of the eddy-parameterized simulations
(1° resolution). A notable point is that after year 0, the residual drifts (CTL) are much smaller than the climate change





trends (CC), which allows us to draw conclusions on the effect of the climate change forcing despite the fact that the spin-up is not yet fully achieved.

It is noteworthy that this experimental design implies that each climate change simulation starts from a different initial state. This strategy has the advantage that each initial state represents an equilibrated control state for the given set of resolution and parameter choice. For each simulation, the impact of climate change is evaluated as the difference between the climate change and preindustrial control simulations, and this difference depends on model choices but may also depend on the differences in the control simulations. Another option would have been to start the climate change simulations from the same initial conditions but this would have had the consequences of a strong drift at the start of the eddy-resolving climate change simulations (as can be seen in the first 40 years of the spin-up, Fig. 1). Our choice is also consistent with the fact that shaping different equilibrium states is part of the role of the eddies. We can also note that none of the eddy-parameterized configurations that we explored has achieved an equilibrium states that comes close to the eddy-resolving configurations (Fig. A1).

## 2.2 Equilibrium states

As noted above, the dynamical and biogeochemical equilibrium states differ for each model configuration. These differences have been more extensively discussed in a similar set up in Lévy et al. (2010, 2012a); Lévy and Martin (2013) and are briefly discussed in the following. It can be noted that the differences between the coarse resolution equilibrium states on one hand, and the fine resolution equilibrium states on the other had, are much larger than the differences within the five (resp. two) coarse resolution (resp. fine resolution) spin-up (Fig. 1).

The main features of the model's solution comprise a western boundary current separating an oligotrophic subtropical gyre in the South of the domain, from a productive subpolar gyre in the North (Fig. 2 and A2). As resolution increases, mesoscale eddies and filamentary structures emerge in the relative surface vorticity field (Fig. 2a, b, c). In all simulations, the barotropic circulation is characterized by an anticyclonic circulation in the south and a cyclonic circulation in the north (Fig. A2). But as resolution increases, non-linear effect become more important and small recirculation gyres appear close to the domain's northern and southern boundaries.

The divergence of the horizontal Ekman flux induces downwelling and a depressed nitracline in the subtropical gyre, as well as upwelling and a raised nitracline in subpolar gyre (Fig. 2 and A4). The model's nitracline follows very closely the thermocline (Fig. A10). Nitrate supplies to the euphotic layer are seasonal, and driven by the seasonal deepening of the mixed-layer, which is strongest in the north of the domain (Fig. A9). This seasonal supply leads to an intense spring bloom in the subpolar gyre and to a weaker winter bloom in the subtropical gyre, where nitrate supplies are lower (Fig. 2 and A2). On an annual average, this leads to a strong contrast between the subpolar gyre, where annual mean NPP is large, and the subtropical gyre characterized by low levels of annual mean NPP. We can also note that nitrate gradients in the thermocline are weaker at fine resolution than at coarse resolution (Fig. 2), in accordance with weaker stratification (Fig. A10). This is associated with lower sub-surface nitrate concentrations. We can note the opposed effect between $1/9°$ and $1/27°$ (Fig. 2) with a slightly larger sub-surface nitrate concentration at $1/9°$ compared to $1/27°$. These differences in equilibrated nitrate distribution lead





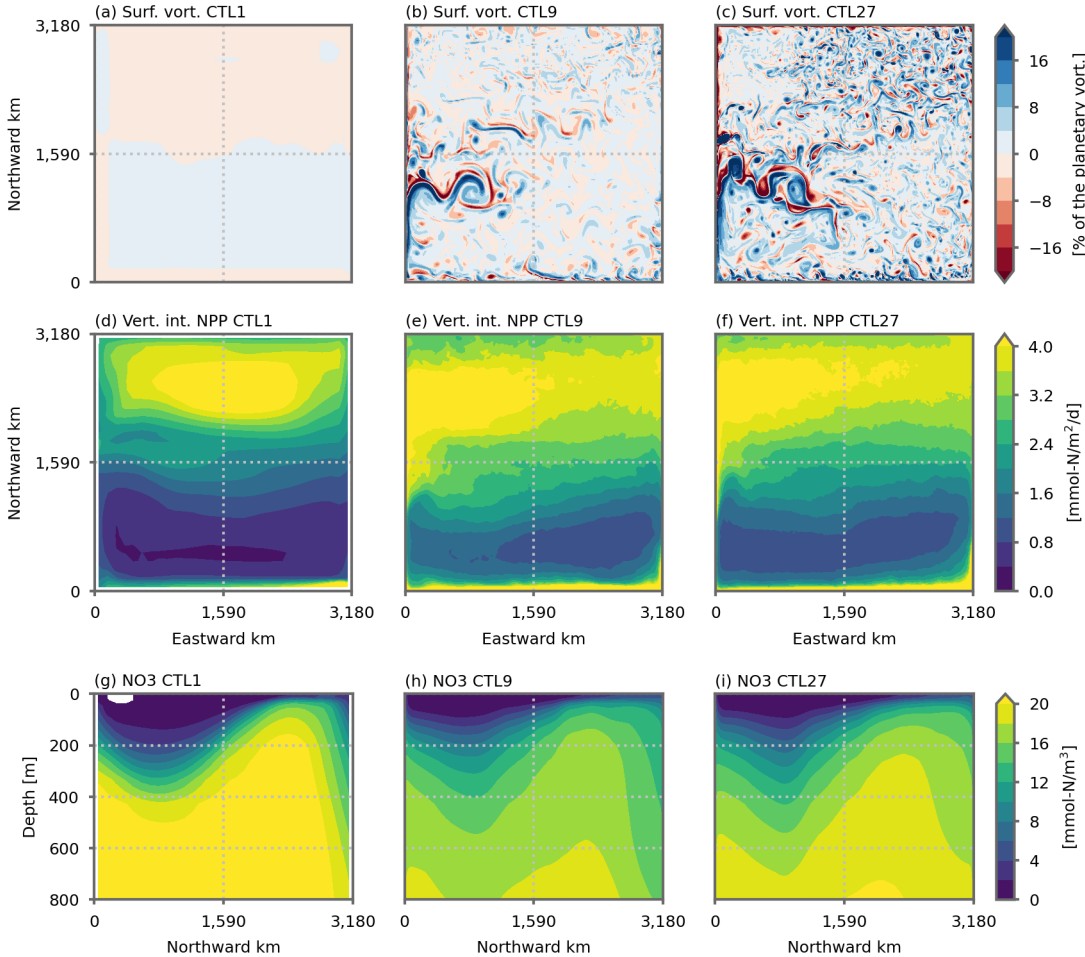

**Figure 2.** (a, b, c) Surface relative vorticity (% of planetary vorticity), (d, e, f) vertically integrated Net Primary Production (NPP, $mmolN.m^{-2}.d^{-1}$) and (g, h, i) vertical distribution of nitrate (zonal mean, $mmolN.m^{-3}$) in the preindustrial control simulations at (a, d, g) $1°$, (b, e, h) $1/9°$ and (c, f, i) $1/27°$ resolution. Vorticity is computed with 2 days averaged velocity fields. NPP and nitrate distribution are averaged over the last five years of the simulations (years 66 to 70). The $1°$ resolution distribution is the mean of the five $1°$ configurations (see Fig. A4 and A2 for individual distributions).

to differences in equilibrated NPP between our different models runs (Fig. 2), but importantly the strong contrast between the two gyres is present is all simulations.

## 2.3 Nitrate budgets

In our simulation, NPP is constrained by nitrate supplies from sub-surface into the sunlit surface layers where photosynthesis can occur. Nitrate budgets are thus used to investigate the processes driving changes in NPP. We focus our analysis on a



predefined region within the model's subpolar gyre where the strongest NPP decline is observed (Fig. 2 and 3). This subpolar box is shown by the black dashed horizontal lines in Fig. 3); it is bounded by the change of sign in the wind stress curl forcing in the South and excludes the northern boundary. The local nitrate budget can be expressed as: $-\boldsymbol{\nabla} \cdot (\boldsymbol{u} \cdot N) + L(N) + \partial_z(k \cdot \partial_z N) + S(N) = \partial_t N$, where $\boldsymbol{\nabla} \cdot (\boldsymbol{u} \cdot N)$ is the divergence of the advective fluxes, $\partial_z(k \cdot \partial_z N)$ is the vertical diffusion term, $L(N)$ is lateral diffusion and $S(N)$ represents the biological nitrate sources (regeneration) and sinks (NPP). $u$ is the

total velocity and includes the bolus velocity of the GM parametrization Gent and McWilliams (1990)) at coarse resolution. Integrating this equation over the subpolar box and depth $D$ leads to:

$$-\underbrace{\oint \boldsymbol{u} \cdot N \, \mathrm{d}s}_{\text{Advective fluxes}} + \underbrace{\int_0^D \langle L(N) \rangle \, \mathrm{d}z}_{\text{Lateral mixing}} + \underbrace{\langle k \cdot \partial_z N \rangle}_{\text{Vertical mixing}} + \underbrace{\int_0^D \langle S(N) \rangle \, \mathrm{d}z}_{\text{Biological sources/sinks}} = \partial_t \left( \int_0^D \langle N \rangle \, \mathrm{d}z \right) \tag{1}$$

The bracket stands for the horizontal integral over the subpolar box. The first term on the left hand side is the integral of the advective fluxes entering/exiting the supolar box between the surface and depth $D$. It is obtained from the volume integral of

the advective fluxes divergence using the Stoke's theorem. The subpolar box being closed at the east and west, it is computed as the sum of three nitrate advective fluxes: 1) the meridional flux entering through the southern boundary of the box between the surface and depth $D$, 2) the vertical flux entering at depth $D$ and 3) the meridional flux going out through the northern boundary of the box between the surface and depth $D$. Of interest here are the advective fluxes supplying nitrate, thus the fluxes entering the subpolar box, ie. the vertical flux and the meridional flux through the southern boundary (Fig. A5). The total transport term

shown in Figure 4 represents these entering fluxes and contains the sum of the vertical flux and of the meridional flux at the southern boundary. It also contains lateral mixing, which is of second order (Fig. A5).

In the preindustrial control simulations, above $\sim 50$ metres depth, nitrate supplies to the subpolar box are dominated by vertical mixing whereas deeper than $\sim 50$ metres they are dominated by the transport term (Fig. 4b). As resolution increases, this separation continues to hold although the respective intensities are modified: advective fluxes are stronger at $1/9^\circ$ and $1/27^\circ$

resolution than at $1^\circ$ resolution (Fig. 4c, d).

The separation between mean and eddy transport in this budget was achieved following the Reynolds averaging method. $u$ and $N$ are broken into a mean component ($\overline{u}$ and $\overline{N}$) and a fluctuating eddy component ($u'$ and $N'$ with $\overline{u'} = 0$ and $\overline{N'} = 0$), leading to:

$$\oint \overline{\boldsymbol{u} \cdot N} \, \mathrm{d}s = \underbrace{\oint \overline{\boldsymbol{u}} \cdot \overline{N} \, \mathrm{d}s}_{\text{Mean}} + \underbrace{\oint \overline{\boldsymbol{u'} \cdot N'} \, \mathrm{d}s}_{\text{Eddy}} \tag{2}$$

The overbar is an averaging operator (in this study a spatio-temporal annual-mean averaging over boxes of $1^\circ$) used to separate the mean from the fluctuating component over a temporal/spatial scale larger than the typical eddy scales. With this operator, we decomposed the total advection term into a mean part including advection by mean velocities ($\oint \overline{\boldsymbol{u}} \cot \overline{N} \, \mathrm{d}s$) and an eddy part ($\oint \overline{\boldsymbol{u'} \cdot N'} \, \mathrm{d}s$), that we computed as the residual between total and mean advection.

This eddy–mean separation is by nature imperfect and leads to mean and eddy components that remain heterogeneous

(Fig. A11) but is nevertheless useful to get an estimate of the contribution of each component. At $1^\circ$ resolution, the explicit



(Reynolds) eddy term is close to zero and we estimated the parameterized eddy component as the sum between the two terms included in the (Gent and McWilliams, 1990) parameterization: advection by the bolus velocity and isopycnal mixing (Fig. 4, A5 and A6).

Finally, the change in nitrate transport in response to global warming is due to changes in nitrate distribution as well as changes in circulation. In order to evaluate these two contributions, we computed offline diagnostics of nitrate advection, using velocities from the preindustrial control simulations ($\boldsymbol{u}_{CTL}$) and nitrate from the climate change simulations ($N_{CC}$) or vice versa ($\boldsymbol{u}_{CC}$ and $N_{CTL}$). More precisely, we used the following decomposition of the change in nitrate advective fluxes, $\Delta(\boldsymbol{u} \cdot N) = \oint \boldsymbol{u}_{CC} \cdot N_{CC}\,\mathrm{ds} - \oint \boldsymbol{u}_{CTL} \cdot N_{CTL}\,\mathrm{ds}$ :

$$\Delta(\boldsymbol{u} \cdot N) = \overbrace{\boldsymbol{u}_{CTL} \cdot \Delta N}^{\Delta \text{Nitrate}} + \underbrace{\Delta \boldsymbol{u} \cdot N_{CTL}}_{\Delta \text{Circulation}} + \overbrace{\Delta \boldsymbol{u} \cdot \Delta N}^{\text{Non-linear } \Delta} \tag{3}$$

with:

$$\boldsymbol{u}_{CTL} \cdot \Delta N = \oint \boldsymbol{u}_{CTL} \cdot N_{CC}\,\mathrm{ds} - \oint \boldsymbol{u}_{CTL} \cdot N_{CC}\,\mathrm{ds} \tag{4}$$

$$\Delta \boldsymbol{u} \cdot N_{CTL} = \oint \boldsymbol{u}_{CC} \cdot N_{CTL}\,\mathrm{ds} - \oint \boldsymbol{u}_{CTL} \cdot N_{CTL}\,\mathrm{ds} \tag{5}$$

$$\Delta \boldsymbol{u} \cdot \Delta N = \Delta(\boldsymbol{u} \cdot N) - \Delta \boldsymbol{u} \cdot N_{CTL} - \boldsymbol{u}_{CTL} \cdot \Delta N \tag{6}$$

## 3 Results

### 3.1 Sensitivity of the decline in Net Primary Production to model resolution

After 70 years of forcing by a linear increase in atmospheric temperature (reaching $+2.8\ ^{\circ}$C, similar in magnitude to the global temperature increase at the end of 21st century in the SSP5-8.5 scenario (Tokarska et al., 2020)), NPP has declined in all model simulations (Fig. 3). The simulated declines are particularly strong in the model's subpolar gyre, a region with elevated initial levels of NPP (Fig. 2) and where we will now focus the analysis. The integrated NPP over the model's subpolar 200 box decreases by $-0.83 \pm 0.03$, $-0.51$ and $-0.44\ \mathrm{mmolN m^{-2}.d^{-1}}$ in our 3 model resolution versions, representing $-26 \pm 1$ %, $-14$ % and $-12$ %, with horizontal resolution set at $1^{\circ}$, $1/9^{\circ}$ and $1/27^{\circ}$, respectively (Fig. 3). Thus they differ markedly between each other, with the decline in NPP halved at the finest resolution compared with the decline projected at coarse resolution. Locally, the intensity of the decline in the coarse resolution model's versions exceeds $-1.5\ \mathrm{mmolN.m^{-2}.d^{-1}}$. The intensity and distribution of these local changes slightly varies among the five coarse resolution configurations. But with finer 205 grid resolution, they are strongly reduced (Fig. 3).

### 3.2 Decline in nutrient supply pathways

The decline in NPP in the model's subpolar box stems from the decline in the supply of nutrients supporting phytoplankton growth, which is associated with the decline in nitrate concentrations in the thermocline (Fig. 4a, e and Fig. A4). Indeed, most




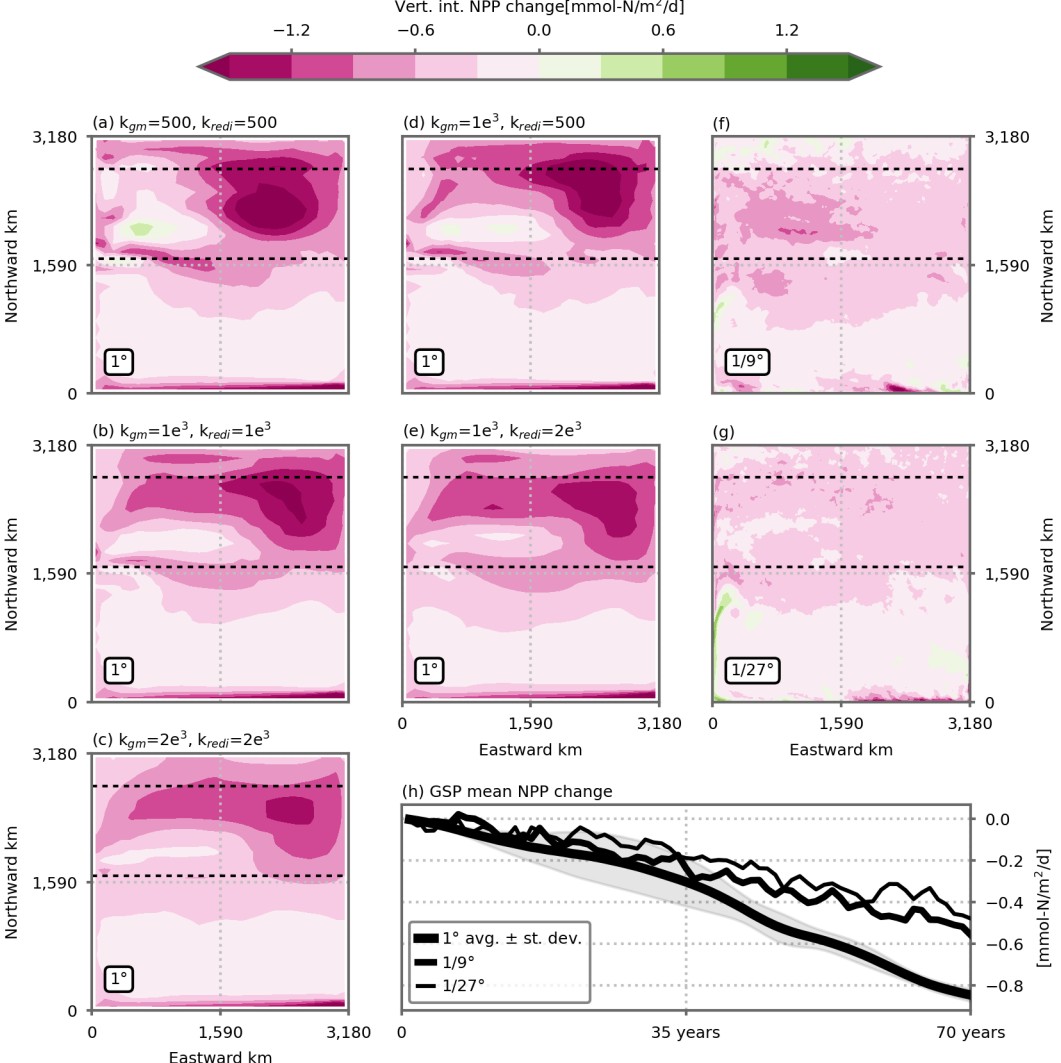

**Figure 3.** Projected decrease in Net Primary Production (NPP, $\mathrm{mmolN.m^{-2}.d^{-1}}$) under a high-emission scenario in (a, b, c, d, e) five $1°$, (f) $1/9°$ and (g) $1/27°$ model simulations. The NPP change is vertically integrated over the entire water column and taken as the difference between the last five years of the climate change and preindustrial control simulations (years 66 to 70). The black dashed horizontal lines delineate the model's subpolar box over which diagnostics for panel (h) and Figures 4, 5, 6 and 7 are performed. This is where simulated NPP is strongest (Fig. 2) and where the simulated NPP decline is also strongest. (h) Time evolution of the NPP decrease in the model's subpolar box in response to climate change, for the three model resolutions. The decrease is computed relative to the preindustrial control simulations. The $1°$ resolution time series shows the average of the five $1°$ configurations. Shading indicates $\pm$ one inter-model standard deviation. See Figure A1 over individual model simulations.


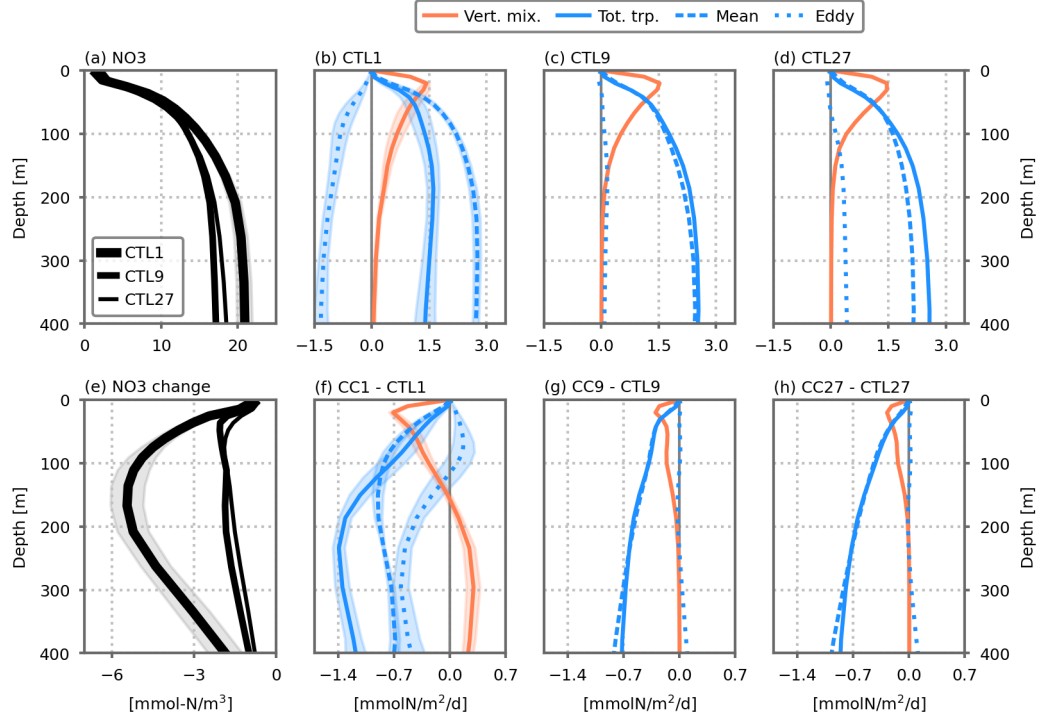

**Figure 4.** Nitrate concentrations $(\mathrm{mmolN.m^{-3}})$ and nitrate supplies $(\mathrm{mmolN.m^{-2}.d^{-1}})$ averaged over the model's subpolar box (black dashed horizontal lines in Fig. 3). (a) Nitrate profiles in the preindustrial control simulations, for the three resolutions. (b, c, d) Dynamical nitrate supplies, cumulated between the surface and depth $D$ in the preindustrial control simulations, against $D$. (e, f, g, h) Changes in these profiles between the climate change and preindustrial control simulations. In the coarse resolution simulations (CTL1/CC1), the profiles show the average of the five $1°$ configurations. Shading indicates $\pm$ one standard deviation. The total transport term (plain blue line) comprises the sum of vertical advection across depth $D$, meridional advection through the southern border of the model's subpolar box; in the case of the $1°$ simulations, total transport also includes eddy parameterizations (including advection by bolus velocity and isopycnal mixing). Total transport is the sum of mean and eddy transport. In the fine resolution simulations (CTL9/CC9, CTL27/CC27), the mean and eddy transport terms are derived from Reynolds decomposition (see the Methodology section). In the coarse resolution simulations (CTL1/CC1), the mean transport represents the resolved transport by the coarse resolution velocity field, and the eddy transport includes the two eddy parameterization terms. See also Figures A5 and A6 for individual fluxes. All profiles are averaged over years 66 to 70.

of the NPP decrease results from a decline in NPP supported by newly supplied nutrients, which are mainly in the form of

nitrate, with the remaining fraction (around 1/3) arising from decreased NPP supported by locally regenerated nutrients (mostly in the form of ammonium, Fig. A3).

A closer examination of nitrate trends within the model's subpolar box provides insights into the physical mechanisms controlling the declining nitrate concentrations in the thermocline in our coarse resolution simulations (Fig. 4). The traditional paradigm (Steinacher et al., 2010; Fu et al., 2016; Bindoff et al., 2019; Kwiatkowski et al., 2020) for the decrease in NPP





is confirmed. Indeed in the euphotic layer (top 150 m), there is a strong reduction in nutrient supply by vertical mixing (Fig. 4f, orange line). This decrease is associated with increasing stratification (Fig. A10). However, similar to what has been shown for the North Atlantic in coarse resolution ESM projections (Whitt, 2019; Tagklis et al., 2020; Whitt and Jansen, 2020), the decrease in nutrient supply is also controlled by the sub-surface nutrient reservoir and the fluxes fueling it. Nitrate concentrations in the top 400 metres strongly decrease under our global-warming scenario (Fig. 4e) because of a reduction in

nutrient advection. Under global warming, the net advective supply of nitrate in the sub-surface (between 100 and 400 metres) is strongly reduced (Fig. 4f, blue line, see also each component of the advective flux in Fig. A6).

   As the resolution increases from 1° to 1/9° and 1/27°, the mechanisms described previously still hold. However, despite similar mean states for the three resolutions at equilibrium (preindustrial control simulations, Fig. 4b, c, d), the climate change-induced decline in vertical mixing to the euphotic layer and in sub-surface advection are both dampened when resolution

increases (Fig. 4f, g, h). At coarse resolution (1°), vertical mixing decreases by $48 \pm 4$ % whereas it only decreases by 18 % in the finest resolution model (Fig. 7). Likewise, the nutrient advection decreases by $83 \pm 5$ % at coarse resolution compared to only 29 % at the finest resolution. A large part of the attenuation occurs when the resolution is changed from 1° to 1/9°, but the attenuation is much more moderate between 1/9° and 1/27°. This is consistent with the expectation that resolution of mesoscale dynamics should be nearly achieved at 1/9° in the latitudinal range considered here, but not fully (see also Fig. 1 and

A1). Moreover, we expect further differences to emerge with grids finer than 1/27° which would allow resolving sub-mesoscale dynamics (Lévy et al., 2010; Gula et al., 2016; McWilliams, 2016; Uchida et al., 2019).

### 3.3    On the role of local eddy processes

   A legitimate question is whether the differences between our coarse resolution and fine resolution models are due to local small-scale eddy processes not adequately captured by the eddy parameterization, or to different responses of the mean large-scale

state to global warming. To address this question, we separated the mean and eddy transport of nitrate, in the two eddy-resolving simulations following Lévy and Martin (2013), with a spatio-temporal annual-mean averaging over boxes of 1°. In the coarse resolution simulations, eddy transport is parameterized by a bolus advection term (Gent and McWilliams, 1990) and isopycnal mixing.

   This analysis reveals the dominant role of the mean advection in fuelling the sub-surface nitrate reservoir at equilibrium in

the two eddy-resolving models (preindustrial control simulations, blue dashed lines in Fig. 4c, d). Eddy advection supplies a minor addition of nutrient in the subpolar box (mostly at 1/27°), essentially through the eddy meridional advection (Fig. A5). In the eddy-parameterized models, the eddy parametrization strongly opposes the resolved transport, removing nutrient from the surface of the subpolar box mainly through the vertical bolus advection term (Fig. 4b and A5).

   In the eddy-resolving climate change simulations, most of the decline in nutrient input is explained by a reduction in mean

nitrate transport (Fig. 4g, h). In the eddy-parameterized simulations, the eddy component plays a larger role and explains $\sim 1/3$ of the total transport decrease at 200 metres (Fig. 4). As for the preindustrial control simulation, it is dominated by changes in the vertical bolus advection term (Fig. A6).

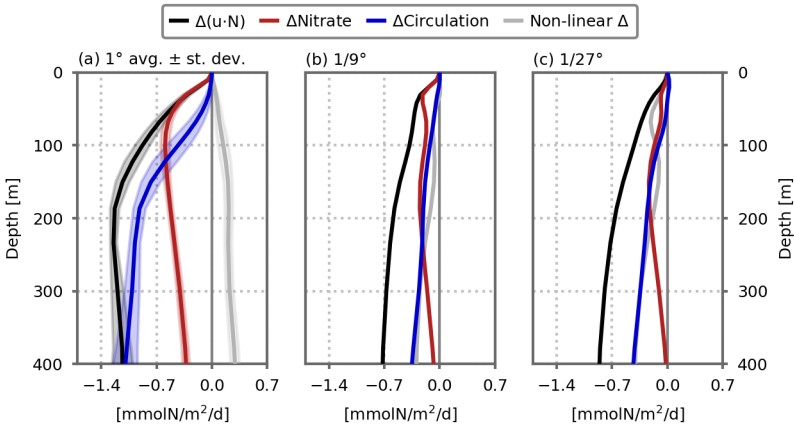

**Figure 5.** Role of circulation change versus nitrate distribution change in explaining the decline of the nitrate advection flux $(\mathrm{mmolN.m^{-2}.d^{-1}})$ in response to global warming in the model's subpolar box, cumulated between the surface and depth $D$, for the (a) $1°$, (b) $1/9°$ and (c) $1/27°$ resolution configurations. The black lines show the simulated change, which is decomposed into three components: the red lines show the component related to the change in nitrate distribution; the blue lines show the component related to the change in circulation; the grey lines show the residual due to non-linear changes. See equations 3 to 6 for details. In (a), the profiles show the average of the five $1°$ configurations. Shading indicates $\pm$ one standard deviation.

### 3.4 On the role of circulation changes

Another question is how changes in nitrate transport (black lines in Fig. 5) are related to changes in circulation. To examine
this question, we first computed in offline mode the advective transport of nitrate keeping the nitrate distribution fixed to its preindustrial state. By doing so, we accounted only for changes in circulation and neglected changes in nitrate distribution (blue lines in Fig. 5). In a similar manner, we kept the circulation constant, and accounted only for the changes in nitrate distribution (red lines in Fig. 5). Due to non-linearities, the sum of circulation-induced (blue lines) and nitrate-induced (red lines) changes is not equal to the total change (black lines) and a residual remains (grey lines in Fig. 5).

In the euphotic layer (upper 0–100 m), the change in nitrate transport is mostly attributable to a change in nitrate concentrations, at all resolutions. But deeper in the water column, the situation is reversed and the change in circulation is responsible for most of the change. In fact at 400 metres depth, the change in nitrate transport is largely explained by circulation changes in all of our simulations. This result highlights the crucial role of circulation changes, which directly affect the resupply of the subsurface nitrate reservoir (between 100–400 m), with the consequence of affecting the mean nitrate gradients, particularly
in the upper layers (0–100 m, Fig. 4e), and thus the nitrate fluxes closer to the surface. We can note a stronger contribution of non-linearities in this simple analysis as model resolution increases.

   To further link our results to circulation changes, we then examined how the water transport through the subpolar box was modified in our climate change simulations (Fig. 6). In our model, the main transport pathway to the subpolar box originates





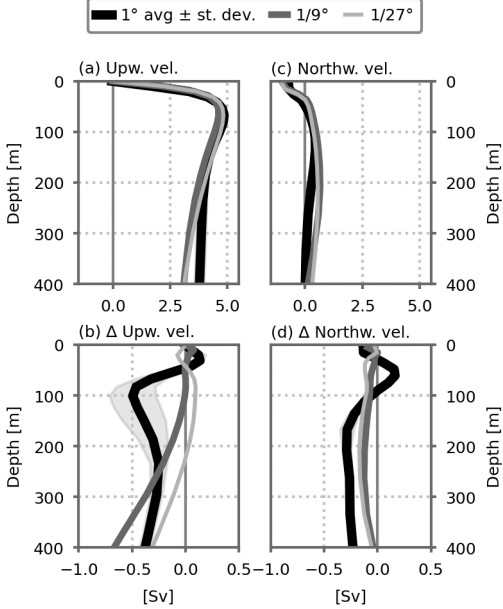

**Figure 6.** Vertical profiles of (a) the northward and (c) upward water transport (in Sv) entering the model's subpolar box (black dashed horizontal lines in Fig. 3) in the preindustrial control simulations for each resolution. (b, d) Changes in these profiles between the climate change and preindustrial control simulations. The thick black lines show the average of the five 1° configurations. Shading indicates ± one standard deviation.

from upwelling and from northward advection at sub-surface (between 50 and 400 m) (Fig. 6a, c). With warming, these upward

and subsurface northward transport both decline (Fig. 6b, d). The mean transport are similar in the equilibrium states between the coarse and fine resolution simulations, but they respond differently to warming. At coarse resolution, the decline of the upward transport is particularly strong, $-0.5$ Sv at $\sim 100$ metres depth (Fig. 6b). This is in phase with the strong decline in nitrate vertical advection at similar depth discussed above (blue lines in Fig. A6). At finer resolution, the decline in the vertical flow is close to zero at $\sim 100$ metres, but increases with depth reaching an amplitude similar to the coarse resolution

simulations at $\sim 400$ metres, $\sim -0.5$ Sv. In the finer resolution simulations, the sub-surface northward flow decline is also weaker, $\sim -0.15$ Sv against $\sim -0.3$ Sv at coarse resolution below 100 metres depth (Fig. 6d).

Previous coarse resolution modelling studies (Whitt, 2019; Tagklis et al., 2020; Whitt and Jansen, 2020) have suggested that the climate-change-driven decline in North Atlantic NPP is strongly linked to changes in the Meridional Overturning Circulation (MOC). This link presumes that the slowdown in the upper branch of the MOC is associated with a decrease in the

strength of the portion of the Gulf Stream that conveys nutrients northward (Whitt, 2019). In our simplified model configuration, the MOC (here diagnosed by the meridional stream function in z-coordinate, Fig. A8) transports waters from the subtropical gyre to the north at depth between $\sim 100$ and $\sim 400$ metres, $1.75 \pm 0.16$ Sv at 1° resolution, 3.14 Sv at 1/9° and 2.94 Sv at 1/27° (flow through the vertical black lines in Fig. A8). Once these waters reach the northern boundary of the domain, they



are entrained downward by convection to depths of 1000 metres and more (Fig. A9) before they return southward towards the
southern boundary where they are upwelled back to the surface. In our coarse resolution simulations, this overturning cell is
strongly slowed down under global warming, inducing in particular a strong reduction, $-1.79 \pm 0.22$ Sv ($-102 \pm 13$ %), in
the water transport between the two gyres (Fig. A8). The slowing of the overturning circulation is associated with a decrease
in the maximum depth of the mixed-layer (MLD) in the northern part of the domain where convection occurs (Fig. A9). The
maximum MLD shoals from over 2000 metres to $\sim 300$ metres after 70 years of climate change forcing (eg. Fig. A9c). At
finer resolution, the MOC is also slowed down, but this slowdown is less pronounced (Fig. A8). The flow between 100 and 400
metres between the two gyres is reduced by $-0.58$ and $-0.85$ Sv ($-18$ and $-29$ %) at $1/9°$ and $1/27°$ resolution, respectively.
It is also exemplified by the maximum depth of the mixed-layer in the northern part of the domain which shoals less than it
does at coarse resolution, from $\sim 1000$ to $\sim 700$ metres (Fig. A9r, u).

The different responses of the MOC between our coarse and fine resolution simulations can be partly related to the differ-
ences in stratification in the North of the domain. The fine resolution equilibrium states are initially less stratified than the
coarse resolution ones (Fig. A10, first column). As a consequence, surface heating during the transient climate warming for
the fine resolution simulations penetrates deeper into the interior and leads to a weaker increase in stratification (Fig. A10, last
column), and thus a weaker decrease in convection and in meridional circulation.

## 4 Conclusions

Based on a wind and buoyancy driven double-gyre model configuration with an idealistic scenario of global warming, we have
shown that the projected decline in NPP is strongly sensitive to horizontal grid resolution, with a decline which is halved at
eddy resolution: $-12$ % versus $-26 \pm 1$ % at coarse resolution (Fig. 7). Moreover, we have also found that this sensitivity is
much stronger than the sensitivity to a large set of eddy parameterization coefficients ($k_r edi$ and $k_g m$). This result is due to the
much weaker decline in nutrient transport at fine resolution, $-29$ % versus $-83 \pm 5$ % at coarse resolution, combined with a
weaker decrease in vertical mixing, $-18$ % vs. $-48 \pm 4$ % (Fig. 7).

A key result of our study is that the resolution-related uncertainties in the NPP projections stem from the high sensitivity
of mean ocean transport projections to resolution. This sensitivity is currently being debated. Regarding the Atlantic MOC
(AMOC), some studies show weaker decline as resolution increases (Roberts et al. (2004); Delworth et al. (2012), as our model
MOC), whereas other studies show an amplified response (Weijer et al., 2012; Spence et al., 2013). It has been suggested that
the intensity of the AMOC decline may be related to its intensity in the control climate, the stronger the initial AMOC the
stronger the decline (Gregory et al., 2005; Winton et al., 2014). The opposite is found here: at fine resolution, the model's
MOC is stronger in the control climate but the response is dampened in the global warming scenario. The AMOC intensity in
the control climate itself has been shown to be sensitive to resolution but with no consensus yet on the sign of this sensitivity
(Winton et al., 2014; Hirschi et al., 2020). The reason for the sensitivity of control and projected AMOC to resolution remains
unclear and differ between studies. The potential causes mentioned are stronger air-sea interactions at fine resolution (Hirschi

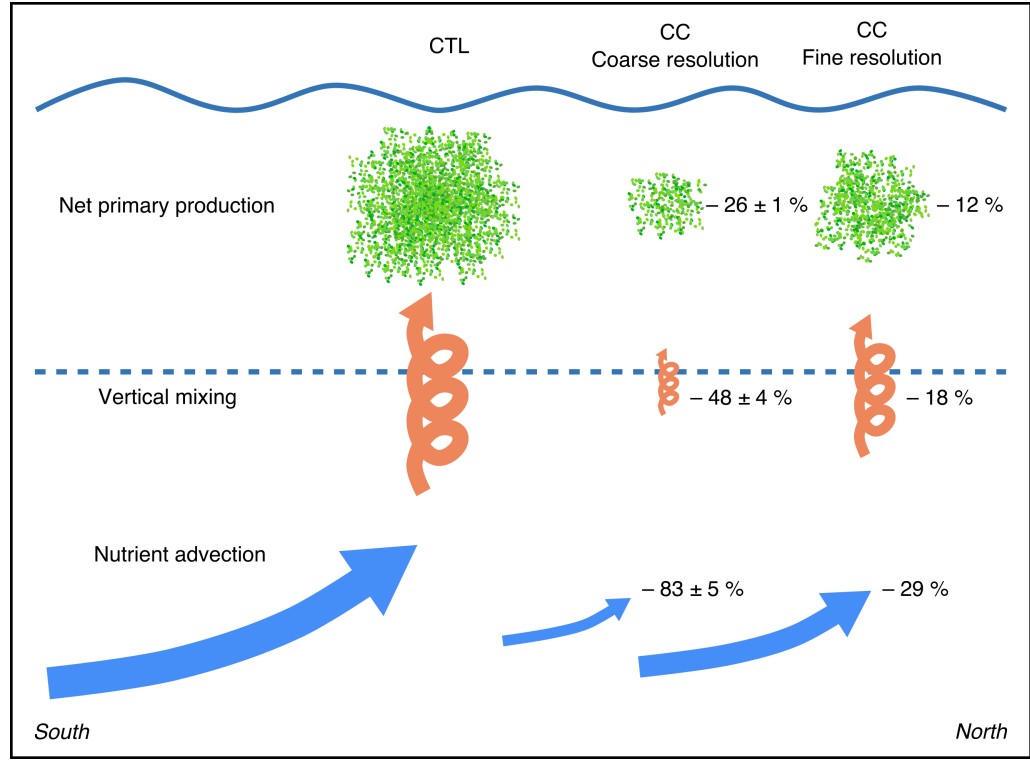

**Figure 7.** Schematic representation of nutrient transport fluxes fueling Net Primary Production (NPP) in the model's subpolar box, and how they are affected by warming in eddy-parameterized coarse (1°) resolution and eddy-resolving fine (1/27°) resolution simulations. The nutrient advection decline is assessed from changes between the surface and 200 metres depth (Total trp. in Fig. 4). Vertical mixing decline is evaluated at 25 metres depth, which is where the decrease is maximum. NPP change is vertically integrated over the entire water column. Changes at coarse resolution are the average of the five 1° configurations ± one standard deviation.

et al., 2020), different spatial distributions of perturbations by the eddies (Spence et al., 2013) or the introduction of biases (Delworth et al., 2012).

Another possible reason for the high sensitivity to resolution of projected ocean circulation changes could be derived from the "eddy saturation" and "eddy compensation" phenomenon which are actively studied in the southern ocean (Farneti et al.,
2010; Farneti and Delworth, 2010; Meredith et al., 2012; Munday et al., 2013; Mak et al., 2017). They stand respectively for the low sensitivity of stratification and overturning circulation to an increase in wind forcing in eddy-resolving models. The connection between the two phenomena is subject to ongoing research (Meredith et al., 2012). Eddy saturation results from the balance between the steepening of the isopycnal by the wind stress and the flattening by eddies (Mak et al., 2017). One could imagine that similar competing processes are at work in our eddy-resolving simulations of climate change, warming
stratifying the ocean versus eddies destratifying it. One clue in that direction is that as in Munday et al. (2013), surface eddy kinetic energy increases with global warming in the eddy-resolving simulations. Eddy saturation is poorly represented by the





GM parametrization (Farneti et al., 2010; Munday et al., 2013, 2014), however simple modification of this parametrization can lead to the emergence of the phenomenon (Mak et al., 2017).

The GM parameterization has improved the representation of circulation in ocean models (Gent, 2011), but its ability to fully
represent the impact of eddies is still questioned (Eden and Greatbatch, 2008; Marshall et al., 2012; Zanna et al., 2017; Mak et al., 2017). The parameterization represents a net sink of energy as the potential energy extracted from the iso-neutral slopes is lost. Moreover, even coupled with laplacian viscosities, it neglects the eddy effects on the momentum equation, the so-called eddy Reynolds stresses (Marshall et al., 2012; Zanna et al., 2017; Mak et al., 2018). These deficiencies in the representation of the eddies have consequences on the mean flows and the transport of tracers (Li et al., 2016; Tamarin et al., 2016; Zanna
et al., 2017; Klocker, 2018). In our simulations, the mean equilibrium state in the eddy-parameterized simulations deviates from the eddy-resolving ones, with little sensitivity to the choice of the GM and lateral diffusion coefficients. And importantly, the global warming driven changes in circulation and the subsequent NPP decline greatly differ from the simulated changes in the eddy-resolving simulations (Fig. 7). This suggests that strong biases in global warming NPP projections could come from inadequate eddy parameterizations. On a positive note, most of the biases stem from differences in the response of the mean
flow to warming, which suggests that a better representations of the eddy impact on the mean flow may correct for it. Ongoing activities in that direction open optimistic perspectives (Marshall et al., 2012; Zanna et al., 2017; Mak et al., 2018) .

The difficulty to attribute recent observed changes in NPP to climate change, and even to detect long-term trends, (Wernand et al., 2013; Beaulieu et al., 2013; Boyce et al., 2014; Henson et al., 2016) make ESMs essential for anticipating anthropogenic changes in NPP. In this regard, it is key to understand the origins and consequences of ESMs biases and/or projection uncer-
tainties. One of such biases, identified here, is the high sensitivity of NPP projections to model resolution. It is questionable how the bias highlighted here in a regional setting can be extrapolated to global ESMs. The oceanic regime closest to our configurations is found in the North Atlantic. Our model captures important characteristics of the North Atlantic NPP system, in particular the leading role of the nutrient advection in refuelling nutrients at sub-surface in the productive subpolar region (Williams et al., 2011), and, of particular interest here, the reduction of NPP in response to global warming (Kwiatkowski
et al., 2020). Global models incorporate additional features such as detailed bathymetry, connections between different basins, response of physiological rates to temperatures changes and ocean-atmosphere feedbacks. Climate change scenarios used in the CMIP6 framework also include complex changes in wind and fresh water fluxes. These elements are likely to further influence the sensitivity of NPP projections to model resolution because of their effects on the Gulf Stream path (Chassignet and Marshall, 2008), the global ocean circulation (Sarmiento et al., 2004; Bronselaer et al., 2016) or on the global warming
driven response of biology (Olonscheck et al., 2013; Lewandowska et al., 2014; Laufkötter et al., 2015) and ocean circulation (Delworth and Zeng, 2008; Weijer et al., 2012; Spence et al., 2013; Bronselaer et al., 2016).

Constraining uncertainties in ESMs NPP projections is crucial because these projections are essential to assess the future evolution of marine biomass and the potential implications for fish catch potential and food security (Lotze et al., 2019). Our results suggest that the uncertainty in NPP decline intensity under global warming may have been underestimated in recent
policy-relevant reports such as IPCC (2019) and IPBES (2019). A natural next step to better assist in political decision making is now to conduct climate change simulation with ESMs better representing oceanic eddies. Two conceivable ways to conduct





such simulations are already taken for ocean dynamics: simulations with ESMs at finer ocean resolution (Haarsma et al., 2016; Gutjahr et al., 2019) and simulations with ESMs using improved eddy parameterizations (Zanna et al., 2017; Mak et al., 2018). Our study calls for pursuing in that direction. Work is in progress to examine the sensitivity of carbon fluxes to eddy resolution. 360 Such advances would be significant for the representation of the global carbon cycle and consequently for climate projections.

*Code availability.* https://github.com/damiencouespel /article_gyre_pp_cc_diagnostics



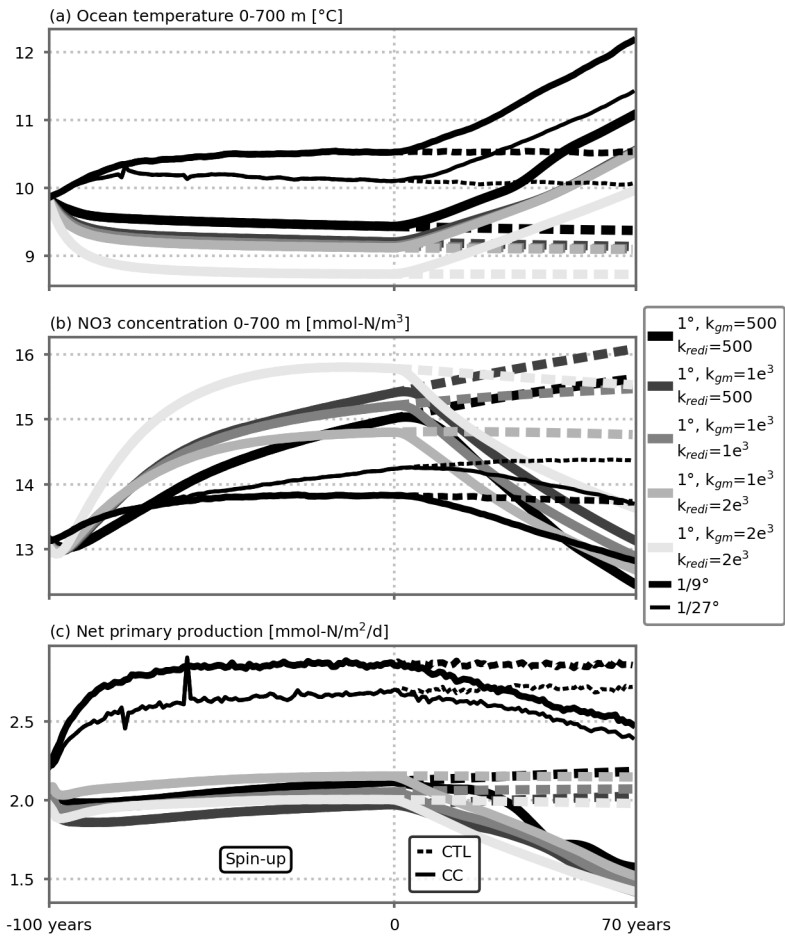

**Figure A1.** Time series of (a) annual ocean temperature ($^\circ$C) and (b) annual nitrate concentration (mmolN.m$^{-3}$) averaged between 0 and 700 metres depth and (c) vertically integrated Net Primary Production (mmolN.m$^{-2}$.d$^{-1}$). All variables are averaged over the whole domain. Years $-100$ to 0 correspond to the spin-up period under seasonal preindustrial forcing. The preindustrial control and climate change simulations discussed in this paper start at year 0. Line thickness stands for the different resolution and the various shades of gray for the various eddy parameterizations. The small peaks in the time series of the $1/27^\circ$ resolution spin-up around years $-80$ and $-60$ are due to a problem during the chaining of the spin-up jobs (which were set by periods of 10 years). These errors did not influence the equilibrium states and we decided not to rerun the spin-up simulations to save computational resources.





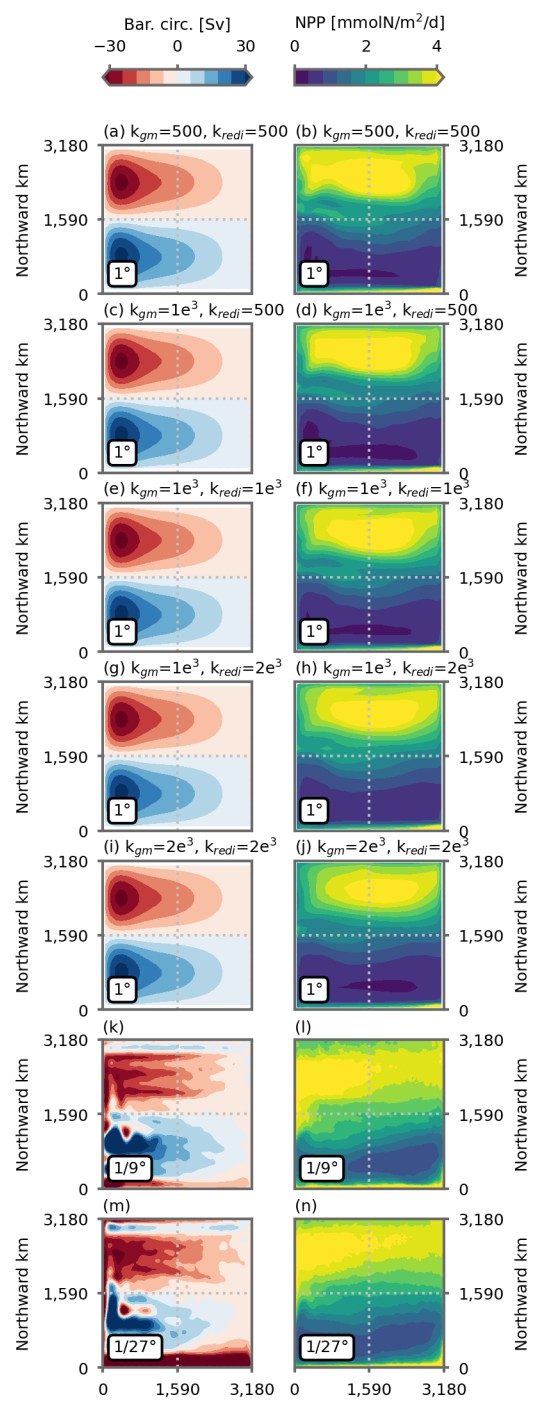

**Figure A2.** (left) Barotropic circulation (Sv) and (right) vertically integrated Net Primary Production (NPP, mmolN.m$^{-2}$.d$^{-1}$) in preindustrial control simulations. The first five rows show the five 1° resolution configurations and the two last one the 1/9° and 1/27° resolution configurations. All figures show average over the last five years of the simulations (years 66 to 70).





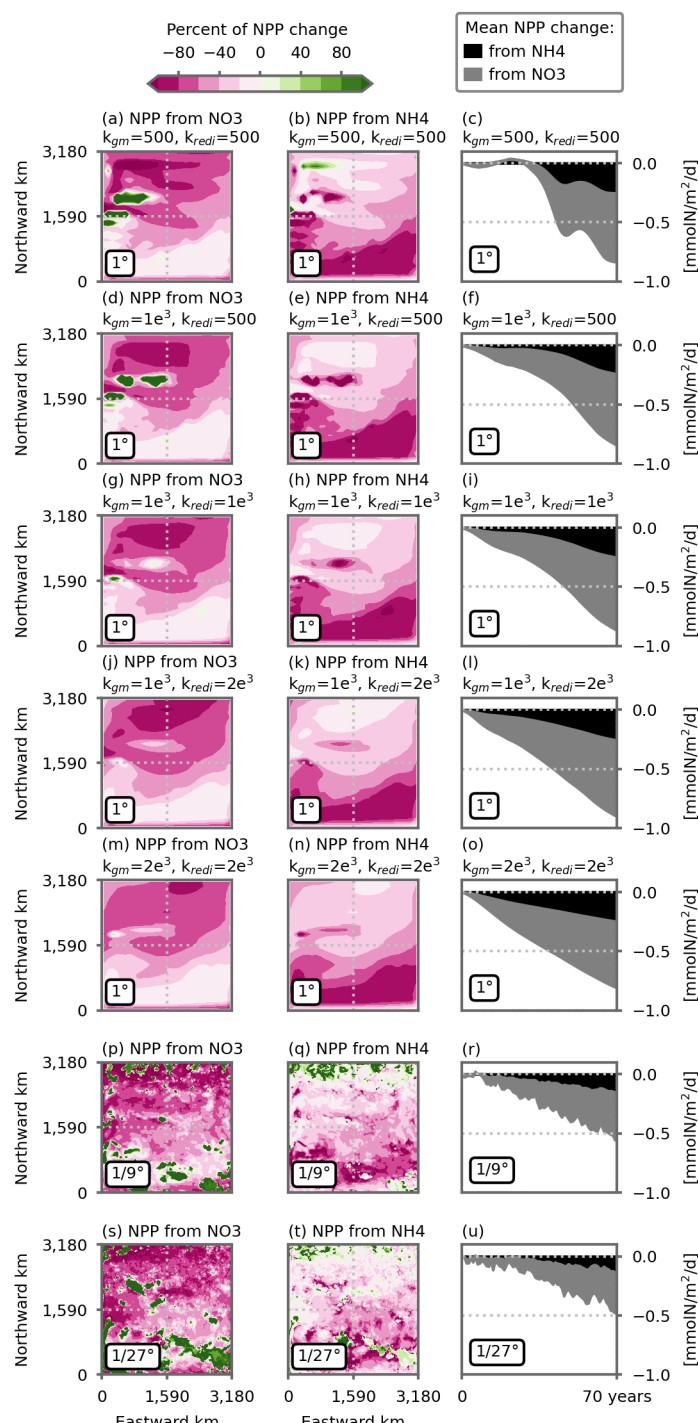

**Figure A3.** Percentage of local change in vertically integrated Net Primary Production (NPP) due to a change in NPP supported by nitrate (first column) and NPP supported by ammonium (second column). Respective share of the NPP decrease from nitrate (black) and ammonium (grey), averaged over the subpolar box (third column, $\mathrm{mmolN.m^{-2}.d^{-1}}$). The first five rows show the five 1° resolution configurations and the two last one the 1/9° and 1/27° resolution configurations. The two first column show average over the last five years of the simulations (years 66 to 70).





**Figure A4.** Vertical distribution of nitrate across the domain (zonal mean, mmolN.m$^{-3}$) in the preindustrial control simulations (first column), the climate change simulations (second column) and difference between the two (third column). The first five rows show the five 1° resolution configurations and the two last one the 1/9° and 1/27° resolution configurations. All figures show average over the last five years of the simulations (years 66 to 70).





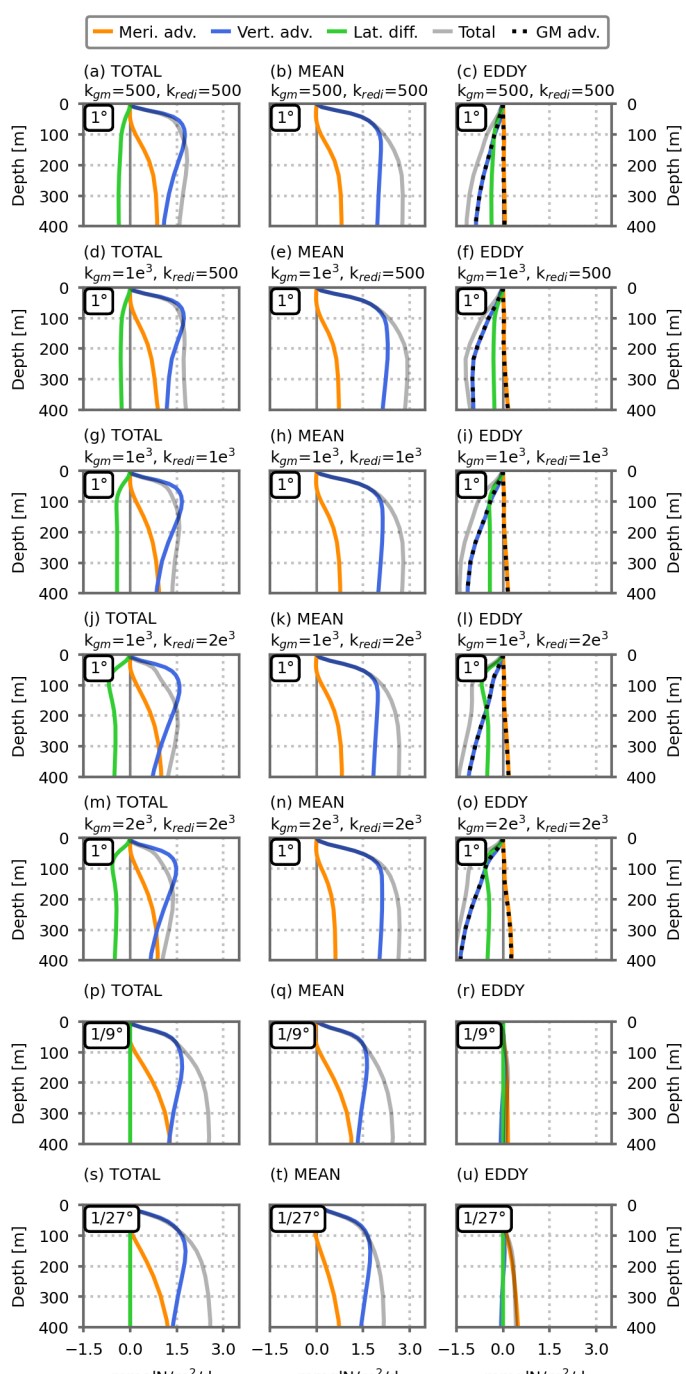

**Figure A5.** Vertical profiles of dynamical nitrate supplies ($\mathrm{mmolN.m^{-2}.d^{-1}}$) in the model's subpolar box (black dashed lines in Fig. 3), cumulated between the surface and depth $D$ in the preindustrial control simulations, against $D$. Total transport (gray lines here, plain blue lines in Fig. 4) is decomposed into 3 components: in orange, the meridional advective flux at the southern border, in blue, the vertical advective flux at depth $D$ and, in green, the lateral mixing. For each resolution, the total advective fluxes (first column) are broken into mean and eddy transport terms (respectively the second and third colums). In the fine resolution simulations ($1/9°$, $1/27°$), they are derived from Reynolds decomposition (see the Methodology section). In the coarse resolution simulations (CTL1/CC1), the mean transport represents the resolved transport by the coarse resolution velocity field, and the eddy transport includes the two eddy parametrizations. The first five rows show the five $1°$ resolution configurations and the two last one the $1/9°$ and $1/27°$ resolution configurations. All figures show average over the last five years of the simulations (years 66 to 70). Positive values indicate an inflow into the subpolar box.




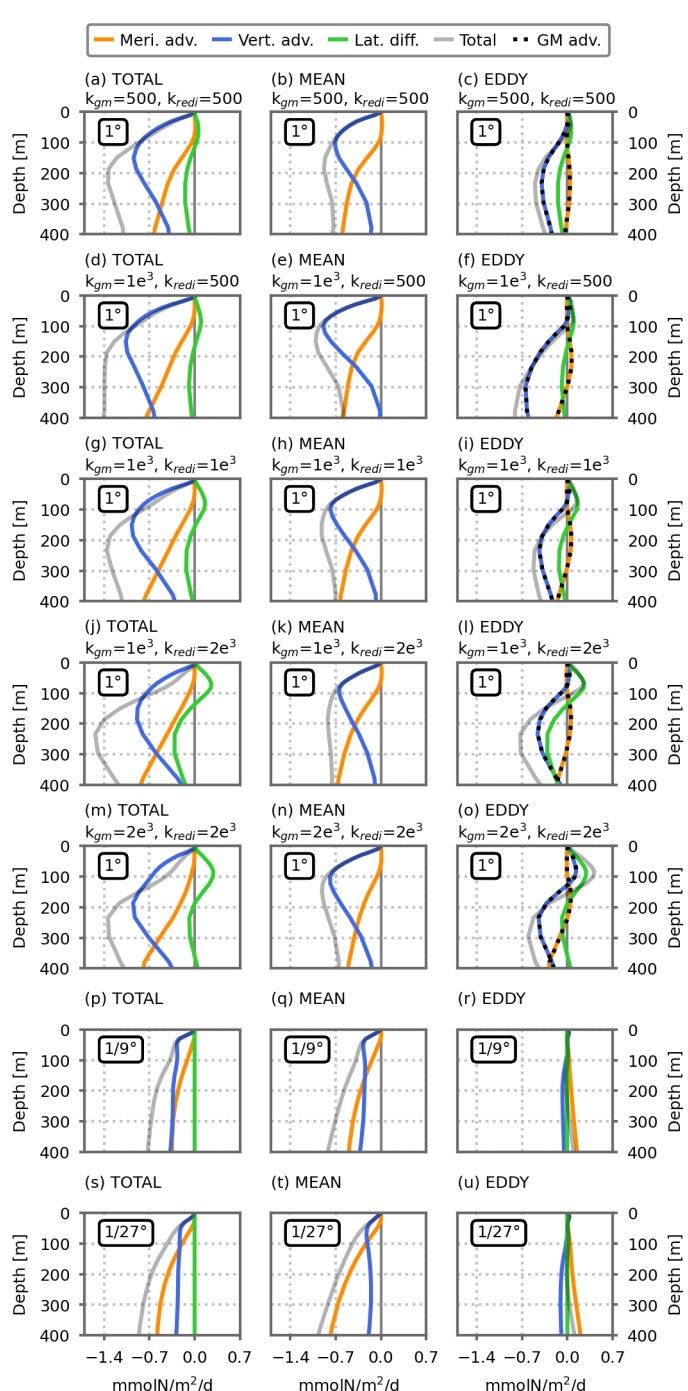

**Figure A6.** Changes in the dynamical nitrate supplies $(\mathrm{mmolN.m^{-2}.d^{-1}})$ between the climate change and preindustrial control simulations. See Figure A5 for details. The change is relative to the preindustrial control simulations. Positive values indicate an increase in nitrate input either because an inflow increases or because an outflow decreases.





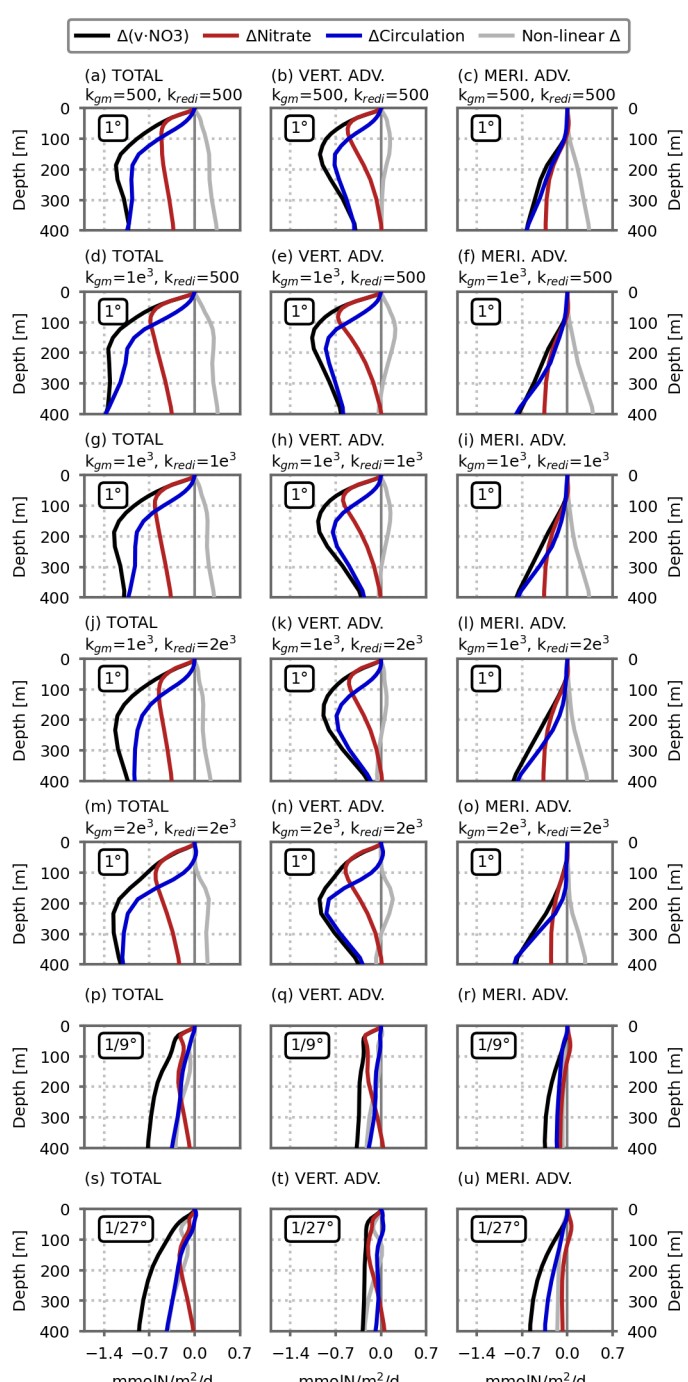

**Figure A7.** Vertical profiles of changes in advective nitrate supplies (black lines, $\mathrm{mmolN.m^{-2}.d^{-1}}$) broken into nitrate distribution change term (red lines), circulation change term (blue lines) and the residual non-linear term (gray lines). Changes are cumulated between the surface and depth $D$ in the model's subpolar box (black dashed lines in Fig. 3) and plotted against $D$. Change is the difference between climate change and preindustrial control simulations For details, see the Methodology section, equations 3 to 6. First column shows total advective fluxes which is the sum of the vertical advective flux at depth $D$ (second column) and the meridional advective flux at the southern border (third column). The first five rows show the five 1° resolution configurations and the two last one the 1/9° and 1/27° resolution configurations. All figures show average over the last five years of the simulations (years 66 to 70).





**Figure A8.** Meridional stream functions (Sv) in the preindustrial control simulation (first column), climate change simulations (second column) and difference between the two (third column). Meridional stream functions is computed in z-coordinates. The vertical black lines range from 100 to 400 metres depth and show the area through which we diagnose the water flow between the two gyres. The first five rows show the five 1° resolution configurations and the two last one the 1/9° and 1/27° resolution configurations. All figures show average over the last five years of the simulations (years 66 to 70).





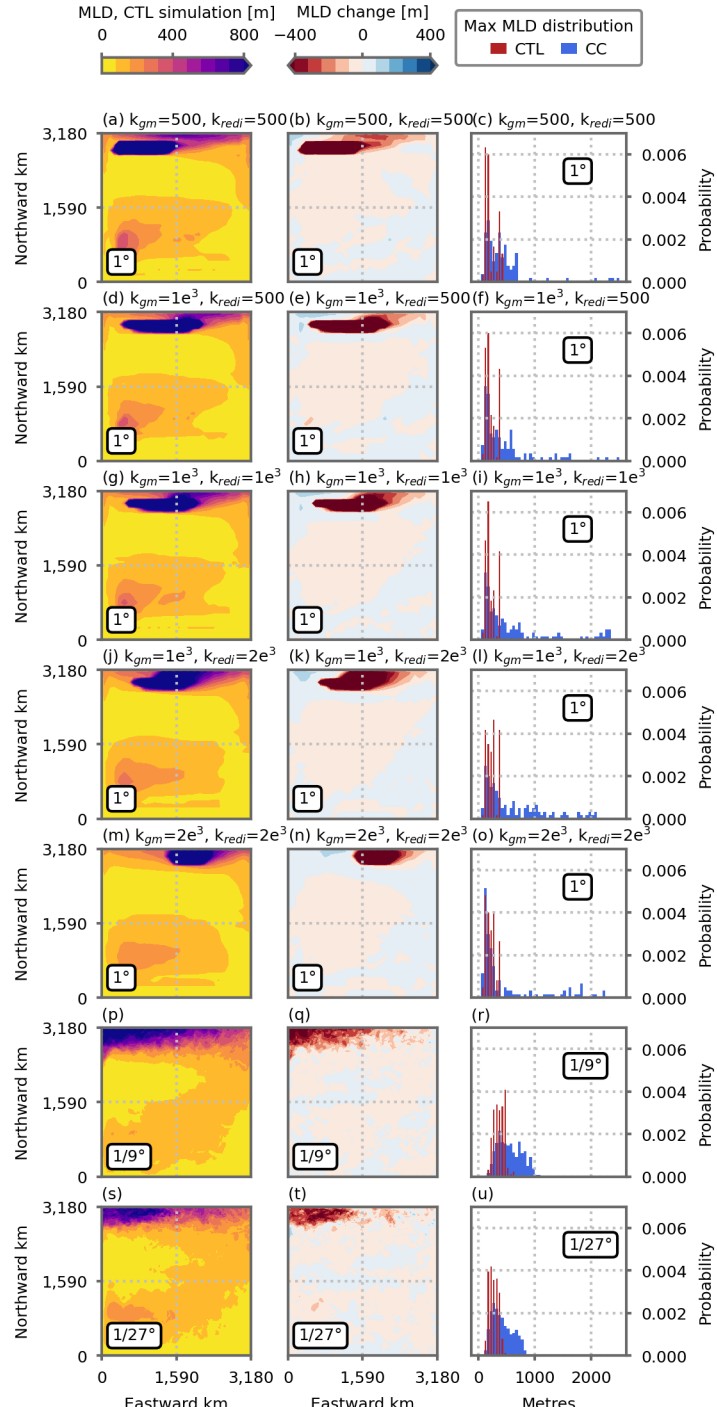

**Figure A9.** Maximum depth of the mixed-layer (metres) reached along the seasonal cycle (monthly averages) in the preindustrial control simulations (first column) and differences in the climate change simulations (second column). (Third column) Distributions of maximum mixed-layer depth north of the model's subpolar box (black dashed lines in Fig. 3) in the preindustrial control simulations (blue bars) and the climate change simulations (red bars). The first five rows show the five 1° resolution configurations and the two last one the 1/9° and 1/27° resolution configurations. All figures show average over the last five years of the simulations (years 66 to 70).





**Figure A10.** Vertical distribution of the Brunt-Vaiasala frequency across the domain (zonal mean, $10^{-5}.s^{-1}$) in the preindustrial control simulations (first column), the climate change simulations (second column) and difference between the two (third column). The first five rows show the five $1°$ resolution configurations and the two last one the $1/9°$ and $1/27°$ resolution configurations. All figures show average over the last five years of the simulations (years 66 to 70).




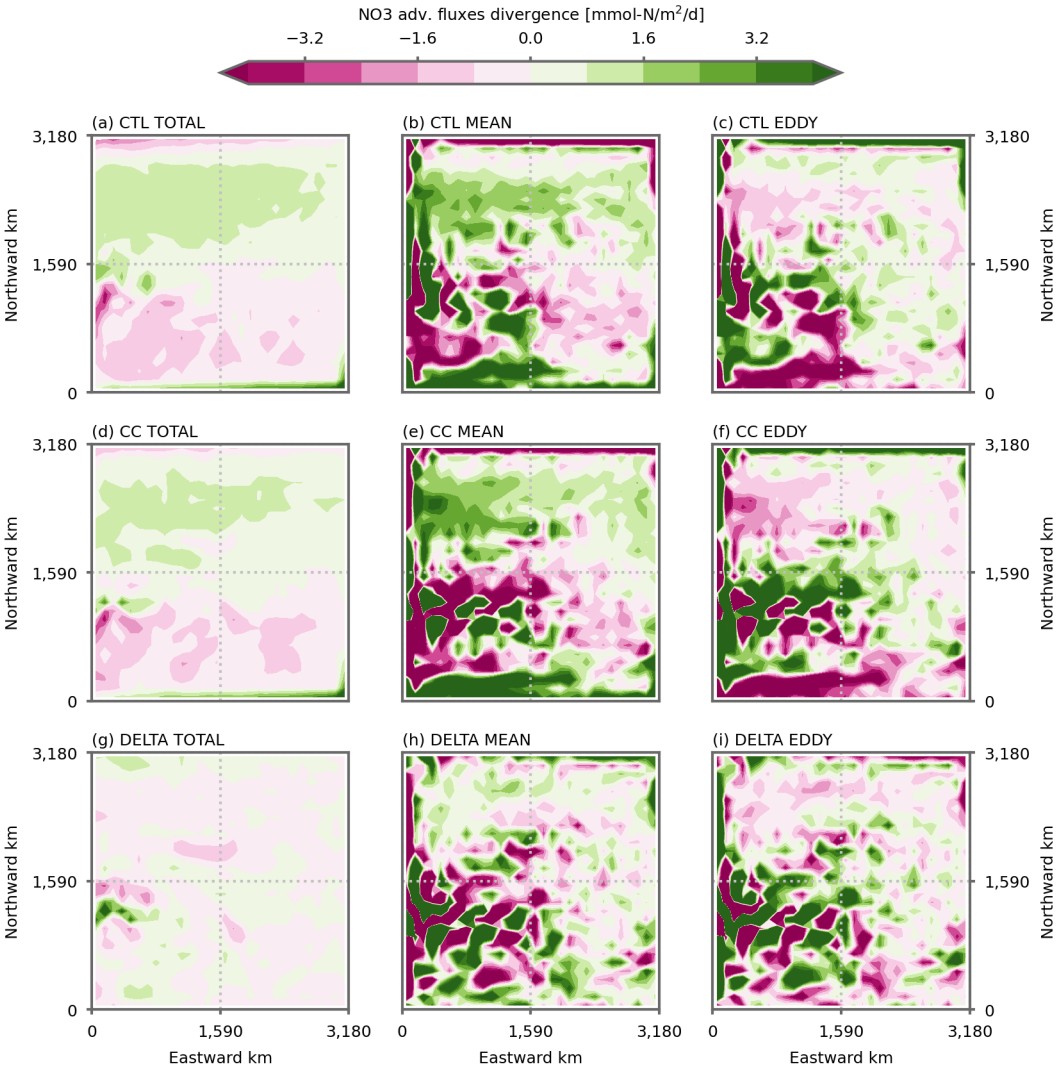

**Figure A11.** Decomposition of the local divergence of the nitrate advective fluxes $(\mathrm{mmolN.m^{-2}.d^{-1}})$ into a mean component and a fluctuating eddy component, for the $1/27°$ simulation. First column, total local divergence, second coulmn, mean component and, third column, residual eddy component. First row, the preindustrial control simulations, second row, the climate change simulations and, third row, the difference between the two. The total component is computed using 2 day averages, the mean component is computed using spatio-temporal ($1°$ and 1 year) average. The eddy component is the residual between the two. The decomposition is integrated vertically between the surface and 400 metres depth and averaged over the last 5 years of each simulations (years 66 to 70). Details of the decomposition in the Methodology section. Positive values indicate an input of nitrate.

*Author contributions.* D.C. wrote the manuscript, conducted the simulations and performed the analysis. M.L. conceived the model experiments. All authors contributed to the writing.





*Competing interests.* The authors declare no competing financial interests.

*Acknowledgements.* This work was supported by ANR project SOBUMS (ANR-16-CE01-0014). This work was granted access to the HPC resources of IDRIS under the allocations 2016-i2016017608, 2016-A0010107608, 2018-A0040107608, 2019-A0070107608 made by GENCI. Support from the European Commission's Horizon 2020 Framework Programme is acknowledged, under Grant Agreement number 641816 for the CRESCENDO project. The authors thank Christian Ethe and Claude Talandier for their help in adapting the model configuration. The authors are grateful for stimulating discussions with Dan Whitt.



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
