# Peer review of "Oceanic primary production decline halved in eddy-resolving simulations of global warming"

_Biogeosciences, 2021_

## Author Comment (AC1)

Dear Scott C. Doney

Thanks again for this positive and helpful review of our manuscript. Please find below a reply to the issues you raised and the way we will address them in the revised version of the manuscript.

Sincerely,

Damien Couespel, Marina Lévy and Laurent Bopp

— — — — — — — — — — — —

*One issue that would be good to address in a little more detail is the difference between the response of new and regenerated production (around Line 210). In the chosen model, ~2/3rds of the NPP decline is due to new production that is directly linked to nitrate supply; previous model studies have indicated substantial variations across models in the temperature sensitivity of NPP under climate warming scenarios that can reflect direct phytoplankton physiological effects as well as changes in the efficiency of nutrient recycling and export (e.g., Laufkötter et al., 2015, doi: 10.5194/bg-12-6955-2015; Laufkötter et al. 2016, Biogeosciences, doi:10.5194/bg-13-4023-2016). It would be useful to know the temperature sensitivity of some of the biological terms in the model, for example. It would also be useful to present briefly some results on the baseline f-ratio in the control simulations and change in f-ratio across the climate change scenario.*

> In our simulations, none of the biogeochemical / biological processes depend on temperature. Thus, the direct effects of warming on phytoplankton physiology are not considered. We agree with the reviewer that this point is a caveat of our study as direct warming effects are key drivers affecting future NPP changes and a large source of uncertainty in explaining contrasted model responses (Laufkötter et al., 2015, Olonschek et al. 2013). Our bias here is to use a very simplistic biogeochemical component, in which all biological terms are independent of temperature. We will emphasize this point in the methods section and in the conclusion (somewhere between lines 337 and 351).

As suggested, we will also compute and discuss the f-ratio and its evolution across the climate change scenario.

*The Results section continues with an analysis of the physical transport differences in the climate response across resolution, linking back to nutrient supply. The issue of changes in circulation is an important aspect of the results. In the discussion of the meridional overturning circulation (MOC) (around line 275) it would be good to clarify the differences in the MOC in this simplified geometry model versus more realistic simulations of the North Atlantic. While the simplified geometry model does include some deep convection at the northern boundary, the overturning circulation is shallow (<1000 m) and weak (only a few Sv in control simulation). Also, in full ESMs, the North Atlantic deep water formation rate and MOC are affected by freshwater export from the Arctic, a process not captured in the simplified model. It would be good to clarify what can be done with the simplified model versus those processes that would require investigation in a more detailed model. A more minor point is that the experiments appear to assume that the seasonal wind stress patterns are constant under climate change, a topic perhaps worth noting in the discussion.*

> We agree that a deeper discussion on the MOC mean-state and projected changes between this simplified geometry and more realistic simulations would be of interest. Roughly, we see 3 major differences that will need to be better discussed in the revised manuscript: 1) closed boundaries (and no topography), 2) no changes in wind stress, 3) no changes in freshwater input.

- Closed boundaries do not allow the inflow of water masses from outside the domain as done in more realistic ocean and climate models:

  - Closed boundaries in the north and in the east prevent any input of water masses from the Arctic and the Nordic Seas. This results in a much simplified temperature, salinity and density vertical profiles (our model only have one homogeneous water mass below 800 metres). In particular Nordic Seas overflows are important for a better representation of the MOC (Zhang et al. 2019), which might be a starting point for explaining the shallower MOC in our simulations.

  - As mentioned by the reviewer, closed boundaries also prevent any input of fresher Arctic waters. The overturning circulation may be more sensitive to changes in these freshwater inputs than changes in precipitations, run-off or ice melting in the considered domain (Bras et al. 2021).

  - Closed boundaries and the simplified geometry may also prevent any latitudinal shift of the Atlantic Meridional Overturning Circulation source regions (Lique et al. 2018).

  - Closed boundary at the southern border of the domain also prevent the simulations from representing all the effects coming from the Southern Ocean.

- As noted, the seasonal-varying  wind stress patterns are held constant under the idealized climate change simulation. Shifts in the wind stress patterns are however important consequences of global warming resulting in a poleward shift of the oligotrophic gyres (Polovina et al. 2008, Yang et al. 2020). However, in the North Atlantic ocean, the impact is weak when compared to temperature changes which are though to be the main drivers of MOC slowdown (Saenko et al. 2005, Gregory et al., 2005; Weaver et al., 2007; Marshall et al., 2015).

As a conclusion, we will add in the discussion section that this simplified model allows to investigate the resolution sensitivity of a warming-induced AMOC decline related with the reduction of the formation of a unique deep water mass - although the link between the two may be more tenuous than previously thought (Lozier et al. 2012). The AMOC response driven by freshwater input and wind stress pattern changes or related to changes in other oceanic regions and water masses would require more realistic configurations.

*Specific issues in text.*

*Line 177*
*"u cot N ds"*
*The "cot" probably used be the command "\cdot" in Latex.*

> Right, this will be rectified

*Line 190*

*In Equation 4, the second N_CC probably should be N_CTL*

> Right, this will be corrected

*Line 298*

*"eddy parameterization coefficients (kredi and kgm)."*

*I think there is a formatting issue here with the subscript. Also, would be good to relate back to terms such as "isopycnal" and "bolus" diffusivity that may be more understandable to the reader rather than model coefficient names, since the specific GM parameterization equations were note presented.*

> Indeed there is a formatting issue, thanks for noting it. As suggested we will used the subscript iso and bol that are more reader friendly.

**References**

Bras, I. L. *et al*. How Much Arctic Fresh Water Participates in the Subpolar Overturning Circulation? *Journal of Physical Oceanography* **51**, 955–973 (2021).

Bronselaer, B., Zanna, L., Munday, D. R. & Lowe, J. The influence of Southern Ocean winds on the North Atlantic carbon sink. *Global Biogeochemical Cycles* **30**, 844–858 (2016).

Gregory, J. M. *et al*. A model intercomparison of changes in the Atlantic thermohaline circulation in response to increasing atmospheric CO 2concentration. *Geophysical Research Letters* **32**, n/a—-n/a (2005).

Laufkötter, C. *et al*. Drivers and uncertainties of future global marine primary production in marine ecosystem models. *Biogeosciences* **12**, 6955–6984 (2015).

Lique, C. & Thomas, M. D. Latitudinal shift of the Atlantic Meridional Overturning Circulation source regions under a warming climate. *Nature Climate Change* **8**, 1013–1020 (2018).

Lozier, M. S. Overturning in the north atlantic. *Annual review of marine science* **4**, 291–315 (2012).

Marshall, J. *et al*. The ocean's role in the transient response of climate to abrupt greenhouse gas forcing. *Clim Dyn* **44**, 2287–2299 (2015).

Olonscheck, D., Hofmann, M., Worm, B. & Schellnhuber, H. J. Decomposing the effects of ocean warming on chlorophyll a concentrations into physically and biologically driven contributions. *Environmental Research Letters* **8**, 14043 (2013).

Polovina, J. J., Howell, E. A. & Abecassis, M. Ocean's least productive waters are expanding. *Geophysical Research Letters* **35**, L03618 (2008).

Saenko, O. A., Fyfe, J. C. & England, M. H. On the response of the oceanic wind-driven circulation to atmospheric CO2 increase. *Climate Dynamics* **25**, 415–426 (2005).

Weaver, A. J., Eby, M., Kienast, M. & Saenko, O. A. Response of the Atlantic meridional overturning circulation to increasing atmospheric CO2: Sensitivity to mean climate state. *Geophysical Research Letters* **34**, (2007).

Yang, H. *et al*. Poleward shift of the major ocean gyres detected in a warming climate. *Geophysical Research Letters* **47**, (2020).

Zhang, R. *et al*. A review of the role of the atlantic meridional overturning circulation in atlantic multidecadal variability and associated climate impacts. *Reviews of Geophysics* **57**, 316–375 (2019).

---

## Author Comment (AC2)

Dear Christopher Sabine,

Thanks for those positive and helpful comments on our manuscript. Please, find below replies to the issues you raised and the way we will address them in the revised version of the manuscript.

Sincerely,

Damien Couespel, Marina Lévy and Laurent Bopp

———————————

*The one aspect that I thought could use a little clarification is how the MOC works in the simplified, two-gyre model. The authors state that the two-gyre model could represent the Atlantic or the Pacific, but of course in the real world the MOC is quite different in these two oceans. I was unclear what exactly drives the MOC in this configuration and how real-world climate change effects that earth system models have suggested will lead to a slowdown of the MOC would be replicated in this two-gyre model.*

> We agree that a clarification of the factors driving the MOC slowdown in our simplified model as compared to more realistic configurations would be of interest. This was also pointed out by the other reviewer. Change in our model's MOC is driven by change in the air-sea heat flux which is thought to be the primary driver of a slowdown of the MOC in climate models (Gregory et al. 2005, Weaver et al. 2007, Marshall et al., 2015). However changes in wind stress, freshwater inputs are recognized to influence the MOC (Bras et al. 2021, Saenko et al. 2005, Bronsaeler et al. 2016, Yang et al. 2020).

In the conclusion we will state that this simplified model allows us to investigate the resolution sensitivity of warming induced AMOC decline related with the reduction of the formation of a unique deep water mass - although the link between the two may be more tenuous than previously thought  (Lozier et al. 2012). AMOC response driven by freshwater input and wind stress pattern changes or related to changes in other oceanic regions and water masses would require more detailed and realistic configurations.

*I would also like to see at least some recognition in the manuscript that this work is examining the climate change effects only on the idealized large-scale open ocean NPP. The simplified two-gyre model with vertical walls and only one ocean, clearly does not reflect the complexities of the real world with dynamic coastal regions and marginal seas that may respond very differently to climate change and anthropogenic forcing. It also does not address how changing ecosystems, for example nitrogen fixers, might take advantage of the increased stratification and reduced nitrogen supply to compensate for a decline in the traditional primary producers. I don't think the lack of coastal waters or multiple ocean basins is a problem, but it should be recognized that this is just one piece of a much broader and more complicated response of the ocean to climate change.*

> We agreed that the manuscript should better emphasize that our work is just one piece of a much broader and complicated response of the ocean to climate change. Thanks for pointed this out. We will emphasize this point in the abstract and the conclusion section. In particular, as mentioned above and in the reply to the other reviewer, we will add in the conclusion that because of the

closed boundaries our model do not allow the inflow of water masses significant for a more realistic MOC.

*I appreciate all the figures in the manuscript and as part of the appendix. The one figure that I did not find particularly interesting or necessary is figure 7. I appreciate that the authors were trying to produce a summary infographic, but this did not clearly convey the idea that model resolution was the driver for the changes outlined in the figure. Perhaps something more than just the words at the top to illustrate this central aspect of the study.*

> We agreed the main results of our study (impact of increasing resolution) may be better emphasized in this figure. We will modify this figure in consequence.

References

Bras, I. L. *et al*. How Much Arctic Fresh Water Participates in the Subpolar Overturning Circulation? *Journal of Physical Oceanography* **51**, 955–973 (2021).

Bronselaer, B., Zanna, L., Munday, D. R. & Lowe, J. The influence of Southern Ocean winds on the North Atlantic carbon sink. *Global Biogeochemical Cycles* **30**, 844–858 (2016).

Gregory, J. M. *et al*. A model intercomparison of changes in the Atlantic thermohaline circulation in response to increasing atmospheric CO2 concentration. *Geophysical Research Letters* **32**, n/a—-n/a (2005).

Lozier, M. S. Overturning in the north atlantic. *Annual review of marine science* **4**, 291–315 (2012).

Marshall, J. *et al*. The ocean's role in the transient response of climate to abrupt greenhouse gas forcing. *Clim Dyn* **44**, 2287–2299 (2015).

Saenko, O. A., Fyfe, J. C. & England, M. H. On the response of the oceanic wind-driven circulation to atmospheric CO2 increase. *Climate Dynamics* **25**, 415–426 (2005).

Weaver, A. J., Eby, M., Kienast, M. & Saenko, O. A. Response of the Atlantic meridional overturning circulation to increasing atmospheric CO2: Sensitivity to mean climate state. *Geophysical Research Letters* **34**, (2007).

Yang, H. *et al*. Poleward shift of the major ocean gyres detected in a warming climate. *Geophysical Research Letters* **47**, (2020).

---

## Author Response (AR1)

Dear Editor,

Please find enclosed a revised version of our manuscript together with a point-by-point response to all reviewer comments and a version of the manuscript with changes highlighted. We thank all the reviewers for their very useful comments that helped to improve the manuscript.

We also made some extra minor corrections to the manuscript: typography, fresh references, small rewordings or clarifications. These corrections all appear in the pdf highlighting the changes.

Sincerely,

Damie Couespel, Marina Lévy and Laurent Bopp

—————————————————————
RC1
—————————————————————

*Oceanic primary production decline halved in eddy-resolving simulations of global warming*

*Damien Couespel, Marina Lévy, and Laurent Bopp*

*Biogeosciences Discussions*

*https://doi.org/10.5194/bg-2021-14*

*The manuscript is well constructed in terms of the scientific numerical experiments, analysis and interpretation of the model output, and presentation. The scientific topic is importance and broadly relevant to the ocean biophysical and biogeochemical research communities. Below, I describe a few areas where the text could be amplified or clarified as well as a few minor specific issues. Overall this is an excellent contribution and should be published after minor revisions.*

*The manuscript brings together two important current lines of ocean science modeling: 1) quantifying the response of ocean productivity to climate change, and 2) characterizing the influence of mesoscale dynamics on phytoplankton growth. The Earth System Models used to project future climate change impacts on marine plankton ecosystems and biogeochemistry are limited computationally to relatively coarse spatial resolution that do not capture mesoscale dynamics. These simulations indicate the decline in global marine primary production, though with considerable cancellation of regional patterns of positive and negative trends and variations across current generation models. The decline in primary production has important implications for marine fisheries and conservation.*

*Previous modeling and field studies indicate the mesoscale features can enhance nutrient supplies in many ocean regions and are thus important for correctly simulating primary production rates and patterns. The lack of these mesoscale biophysical process could bias future climate change projections. The authors of this study conduct novel climate experiments varying spatial resolution in an ocean model with idealized, two-gyre geometry. The model is integrated at eddy-resolving through to eddy-parameterized resolution, with several different versions of the eddy-parameterized simulation using different combinations of horizontal diffusivity in the Gent-McWilliams parameterization.*

*The model description and experimental design subsections of the methodology are solid, detailed and informative, with sufficient details provided for other researchers to replicate the basics of the*

*study. The experimental design includes description of the control simulation, model spin-up for the different experimental cases, and the pre-industrial and climate change integrations.*

*The model analysis is framed the changes in net primary production (NPP) and on a budget of the various physical advective transport and mixing terms regulating the nutrient (nitrate) supply to the surface waters of the subpolar gyre. The nitrate budget analysis is solidly based, building on a number of previous studies analyzing time mean and variability (Reynolds decomposition) of the North Atlantic nitrate budget from the perspective of both vertical and lateral nutient supply terms (e.g., McGillicuddy et al., Global Biogeochemical Cycles, 2003, doi:10.1029/2002GB001987).*

*The climate change simulations in the simplified geometry model exhibit a decline in NPP in the subpolar gyre similar the the results seen Earth System Models for the North Atlantic (e.g., Bopp et al., 2013; Kwiatkowski et al., 2020). The NPP decline is stronger in absolute and fractional terms for the coarse resolution model, and the analysis links those declines to a reduction in nitrate supply. Similar to previous coarse resolution simulations, the model shows declines in nutrient supply both due to increased stratification (reduced supply vertical mixing) and decline in nitrate in the thermocline linked to lateral processes.*

*One issue that would be good to address in a little more detail is the difference between the response of new and regenerated production (around Line 210). In the chosen model, ~2/3rds of the NPP decline is due to new production that is directly linked to nitrate supply; previous model studies have indicated substantial variations across models in the temperature sensitivity of NPP under climate warming scenarios that can reflect direct phytoplankton physiological effects as well as changes in the efficiency of nutrient recycling and export (e.g., Laufkötter et al., 2015, doi: 10.5194/bg-12-6955-2015; Laufkötter et al. 2016, Biogeosciences, doi:10.5194/bg-13-4023-2016). It would be useful to know the temperature sensitivity of some of the biological terms in the model, for example. It would also be useful to present briefly some results on the baseline f-ratio in the control simulations and change in f-ratio across the climate change scenario.*

> In our simulations, none of the biogeochemical / biological processes depend on temperature. Thus, the direct effects of warming on phytoplankton physiology are not considered. We agree with the reviewer that this point is a caveat of our study as direct warming effects are key drivers affecting future NPP changes and a large source of uncertainty in explaining contrasted model responses (Laufkötter et al., 2015, Olonschek et al. 2013). Our bias here is to use a very simplistic biogeochemical component, in which all biological terms are independent of temperature.

This is now clearly stated:

- in the methodology section, lines 73-74: « *Note that in the LOBSTER model, none of the biological rate are directly dependent on temperature, allowing us to focus our analysis on effects due to changes in physical transport.* »

- in the results section, lines 215-220: « *The mean f-ratio on the subpolar box is about 0.43 all along the simulations (Fig. A3, fourth column) albeit a very small decline (from ~ 0.43 to ~ 0.40) in the climate change simulations. This slight decline is consistent with the lower primary production regime having lower f-ratio (Sarmiento and Gruber, 2006, p. 166). We should also note that changes in f-ratio in response to warming might be underestimated here because the direct impact of increasing temperature on biological rates, which has been shown to affect the response of biology to climate change (Olonscheck et al., 2013; Lewandowska et al., 2014; Laufkötter et al., 2015), is not accounted for in this study.* »

- in the conclusion section, lines 362-364: « *Global ESMs and CMIP6 framework scenarios incorporate such additional features, as well as the temperature sensitivity of biological processes that has been revealed to also affect the global warming driven response of biology. (Olonscheck et al., 2013; Lewandowska et al., 2014; Laufkötter et al., 2015)* ». Note that this paragraph (lines 346-365) was fully re-written (see below).

As suggested, the f-ratio was computed (Fig. A3, last column) and briefly discussed as mentioned above. Caption of figure A3 now reads:

« *Figure A3. Percentage of local change in vertically integrated Net Primary Production (NPP) due to a change in new NPP (New NPP, first column) and regenerated NPP (Reg NPP, second column). Respective share of the NPP decrease from New NPP (black) and Reg NPP (grey), averaged over the subpolar box (third column, mmolN.m–2.d–1). Evolution of the f-ratio averaged over the subpolar box in the preindustrial control and climate change simulations (fourth column, respectively dashed and plain lines). Vertical integration up to 100 metres depth where most of the NPP occurs. New NPP is the NPP supported by nitrate minus nitrification and Reg NPP is NPP supported by ammonium plus nitrification. The nitrification is handled in this way so that locally produced nitrate is counted as a source of regenerated production and not as a source of new production. The f-ratio is computed as the ratio between New NPP and total NPP. The first five rows show the five 1° resolution configurations and the two last one the 1/9° and 1/27° resolution configurations. The two first column show average over the last five years of the simulations (years 66 to 70).* »

*The Results section continues with an analysis of the physical transport differences in the climate response across resolution, linking back to nutrient supply. The issue of changes in circulation is an important aspect of the results. In the discussion of the meridional overturning circulation (MOC) (around line 275) it would be good to clarify the differences in the MOC in this simplified geometry model versus more realistic simulations of the North Atlantic. While the simplified geometry model does include some deep convection at the northern boundary, the overturning circulation is shallow (<1000 m) and weak (only a few Sv in control simulation). Also, in full ESMs, the North Atlantic deep water formation rate and MOC are affected by freshwater export from the Arctic, a process not captured in the simplified model. It would be good to clarify what can be done with the simplified model versus those processes that would require investigation in a more detailed model. A more minor point is that the experiments appear to assume that the seasonal wind stress patterns are constant under climate change, a topic perhaps worth noting in the discussion.*

> We agree that a deeper discussion on the MOC mean-state and projected changes between this simplified geometry and more realistic simulations would be of interest. In the conclusion section, the paragraph between lines 346-365 was fully rewritten to account for that comment and and to clearly state what can and cannot be done with our model.

The paragraph now reads:

« *The difficulty to attribute recent observed changes in NPP to climate change, and even to detect long-term trends (Wernand et al., 2013; Beaulieu et al., 2013; Boyce et al., 2010; Henson et al., 2016), make ESMs essential for anticipating anthropogenic changes in NPP. In this regard, it is key to understand the origins and consequences of ESMs biases and/or projection uncertainties. One of such biases, identified here, is the high sensitivity of NPP projections to model resolution. It is questionable how the bias highlighted here in a idealized regional setting can be extrapolated to global ESMs. The oceanic regime closest to our configurations is found in the North Atlantic. Our*

*model captures important characteristics of the North Atlantic NPP system, in particular the reduction of NPP in response to global warming (Kwiatkowski et al., 2020) as well as the leading role of the nutrient advection in refuelling nutrients at sub-surface in the productive subpolar region (Williams et al., 2011). With our simplified model, we were able to investigate the consequences of a decline in the MOC driven by changes in air-sea heat fluxes and associated with the reduction in deep water formation of a unique water mass. However, changes in the AMOC may also be driven by freshwater input (Bras et al., 2021), or by changes in wind stress pattern (Polovina et al., 2008; Yang et al., 2020), or related to changes in adjacent regions and involving the formation of different water masses (Delworth and Zeng, 2008; Bronselaer et al., 2016; Lique and Thomas, 2018; Bras et al., 2021). Moreover, the link between the reduction in deep water formation at high latitudes in the North Atlantic and the slowing of the AMOC may be more tenuous than previously thought (Lozier, 2012). Investigation of these different aspects would require more realistic configurations and more complex climate change scenarios. Global ESMs and CMIP6 framework scenarios incorporate such additional features, as well as the temperature sensitivity of biological processes that has been revealed to also affect the global warming driven response of biology. (Olonscheck et al., 2013; Lewandowska et al., 2014; Laufkötter et al., 2015). All these elements are likely to further influence the sensitivity of NPP projections to model resolution. »*

*Specific issues in text.*

*Line 177*

*"u cot N ds » The "cot" probably used be the command "\cdot" in Latex.*

> done

*Line 190*

*In Equation 4, the second N_CC probably should be N_CTL*

> done

*Line 298*

*« eddy parameterization coefficients (kredi and kgm). »*

*I think there is a formatting issue here with the subscript. Also, would be good to relate back to terms such as "isopycnal" and "bolus" diffusivity that may be more understandable to the reader rather than model coefficient names, since the specific GM parameterization equations were note presented.*

> Done in the entire manuscript. We also have changed the term « lateral diffusion » to « isopycnal diffusion » (or « lateral mixing » to « isopycnal mixing ») in the whole manuscript to be consistent. The term « isopycnal diffusion » is not appropriate for the 1/27° resolution simulation (diffusion is along the horizontal). However because diffusion in the 1/27° simulations is minimal and only to insure numerical stability we decided to keep the term « isopycnal ». This is emphasized:

- in the methodology section, lines 85-87: « *Note that diffusion is along the isopycnals only for the 1○ coarse resolution. In the following, we will nevertheless use the term isopycnal diffusion regardless of the resolution for simplicity and because the diffusion is null or minimal at finer resolution.* »

- in the caption of table 1: « *This bilaplacian diffusion acts on the horizontal unlike the laplacian diffusion acting along isopycnals in the 1° resolution simulations.* »

*I found this to be a well thought out scientific investigation into the effects of model resolution on NPP changes under a climate change scenario. The manuscript is well organized and the conclusions are clear. This work clearly illustrates the importance of mesoscale processes on nutrient supply and the limitations of the sub-grid scale parameterizations in coarse resolution models. As an observational oceanographer, the methods description and results generally seemed clear and complete but I defer to other reviewers that are more familiar with model details to judge that.*

*The one aspect that I thought could use a little clarification is how the MOC works in the simplified, two-gyre model. The authors state that the two-gyre model could represent the Atlantic or the Pacific, but of course in the real world the MOC is quite different in these two oceans. I was unclear what exactly drives the MOC in this configuration and how real-world climate change effects that earth system models have suggested will lead to a slowdown of the MOC would be replicated in this two-gyre model.*

> We agree that a clarification of the factors driving the MOC slowdown in our simplified model as compared to more realistic configurations would be of interest. In the conclusion section, the paragraph between lines 346-365 was fully rewritten to account for that comment.

The paragraph now reads:

*« The difficulty to attribute recent observed changes in NPP to climate change, and even to detect long-term trends (Wernand et al., 2013; Beaulieu et al., 2013; Boyce et al., 2010; Henson et al., 2016), make ESMs essential for anticipating anthropogenic changes in NPP. In this regard, it is key to understand the origins and consequences of ESMs biases and/or projection uncertainties. One of such biases, identified here, is the high sensitivity of NPP projections to model resolution. It is questionable how the bias highlighted here in a idealized regional setting can be extrapolated to global ESMs. The oceanic regime closest to our configurations is found in the North Atlantic. Our model captures important characteristics of the North Atlantic NPP system, in particular the reduction of NPP in response to global warming (Kwiatkowski et al., 2020) as well as the leading role of the nutrient advection in refuelling nutrients at sub-surface in the productive subpolar region (Williams et al., 2011). With our simplified model, we were able to investigate the consequences of a decline in the MOC driven by changes in air-sea heat fluxes and associated with the reduction in deep water formation of a unique water mass. However, changes in the AMOC may also be driven by freshwater input (Bras et al., 2021), or by changes in wind stress pattern (Polovina et al., 2008; Yang et al., 2020), or related to changes in adjacent regions and involving the formation of different water masses (Delworth and Zeng, 2008; Bronselaer et al., 2016; Lique and Thomas, 2018; Bras et al., 2021). Moreover, the link between the reduction in deep water formation at high latitudes in the North Atlantic and the slowing of the AMOC may be more tenuous than previously thought (Lozier, 2012). Investigation of these different aspects would require more realistic configurations and more complex climate change scenarios. Global ESMs and CMIP6 framework scenarios incorporate such additional features, as well as the temperature sensitivity of biological processes that has been revealed to also affect the global warming driven response of biology. (Olonscheck et al., 2013; Lewandowska et al., 2014; Laufkötter et al., 2015). All these elements are likely to further influence the sensitivity of NPP projections to model resolution. »*

*I would also like to see at least some recognition in the manuscript that this work is examining the climate change effects only on the idealized large-scale open ocean NPP. The simplified two-gyre model with vertical walls and only one ocean, clearly does not reflect the complexities of the real world with dynamic coastal regions and marginal seas that may respond very differently to climate change and anthropogenic forcing. It also does not address how changing ecosystems, for example nitrogen fixers, might take advantage of the increased stratification and reduced nitrogen supply to compensate for a decline in the traditional primary producers. I don't think the lack of coastal waters or multiple ocean basins is a problem, but it should be recognized that this is just one piece of a much broader and more complicated response of the ocean to climate change.*

> We agreed that the manuscript should better emphasize that our work is just one piece of a much broader and complicated response of the ocean to climate change. We thank the reviewer for pointing this out and for providing us with wording in the statement that we have taken the liberty of reusing. This is now emphasized:

- in the abstract, lines 12-15: « *Although being only one piece of a much broader and more complicated response of the ocean to climate change, our results call for improved representation of the role of eddies on nutrient transport below the seasonal mixed-layer to better constrain the future evolution of marine biomass and fish catch potential.* »

- in the conclusion section, lines 368-370: « *Although being only one piece of a much broader and more complicated response of the ocean to climate change, our results suggest that the uncertainty in NPP decline intensity under global warming may have been underestimated in recent policy-relevant reports such as IPCC (2019) and IPBES (2019).* »

*I appreciate all the figures in the manuscript and as part of the appendix. The one figure that I did not find particularly interesting or necessary is figure 7. I appreciate that the authors were trying to produce a summary infographic, but this did not clearly convey the idea that model resolution was the driver for the changes outlined in the figure. Perhaps something more than just the words at the top to illustrate this central aspect of the study.*

> We agreed the main results of our study (impact of increasing resolution) may be better emphasized in this figure. The figure 7 is now:

[Figure]

and the caption now reads:

*« Schematic representation of the projected response of Net Primary Production (NPP) and of the nutrient fluxes (advection and vertical mixing) supporting it to warming, with our (a) eddy-parameterized coarse (1°) resolution and (b) eddy-resolving fine (1/9°, 1/27°) resolution simulations, in the model's subpolar gyre. Decline in nutrient advection is assessed from changes between the surface and 200 metres depth (Total trp. in Fig. 4). Decline in vertical mixing is evaluated at 25 metres depth, which is where the decrease is maximum. NPP change is vertically integrated over the entire water column. Changes at coarse resolution are the average of the five 1° configurations ± one standard deviation. Changes at fine resolution are the average between 1/9° and 1/27° configurations ± half of the difference between the two. The background diagrams in grey represent the initial control situation while the colored diagrams represent the situation after 70 years of warming. »*

*I recommend publication of this manuscript after minor revisions.*

---

## Author Response (AR2)

Dear Editor,

Please find enclosed a slightly revised version of our manuscript. It incorporates the recommended corrections: removing periods in units, inserting spaces between « mmol » and « N » and using subsrcipt « 3 » in « NO3. Thanks for pointing these technical corrections.

Sincerely,

Damien Couespel, Marina Lévy and Laurent Bopp